# Aβ-driven nuclear pore complex dysfunction alters activation of necroptosis proteins in a mouse model of Alzheimer's disease

Vibhavari Aysha Bansal[1], Jia Min Tan[1,2], Hui Rong Soon[1,2], Norliyana Zainolabidin[1], Takaomi Saido[3], Toh Hean Ch'ng[1,2]*

[1]Lee Kong Chian School of Medicine, Nanyang Technological University, Singapore, Singapore; [2]School of Biological Science, Nanyang Technological University, Singapore, Singapore; [3]Department of Neurocognitive Science, Institute of Brain Science, Nagoya City University Graduate School of Medical Sciences, Nagoya, Japan

*For correspondence:
thchng@ntu.edu.sg

Competing interest: The authors declare that no competing interests exist.

## eLife assessment

This study focuses on nuclear pore complex dysfunction in a mouse model of Alzheimer's disease related Aβ pathology. If future revisions can adequately respond to the reviewer comments, the findings may eventually be **useful** in supporting the idea that nuclear cytoplasmic transport defects occur prior to plaque deposition in this disease model and may be caused by Alzheimer's disease pathology. However, even after revision, the work suffers from overinterpretation of some of the data and remains **incomplete** in several respects.

**Abstract** The emergence of Aβ pathology is one of the hallmarks of Alzheimer's disease (AD), but the mechanisms and impact of Aβ in progression of the disease is unclear. The nuclear pore complex (NPC) is a multi-protein assembly in mammalian cells that regulates movement of macromolecules across the nuclear envelope; its function is shown to undergo age-dependent decline during normal aging and is also impaired in multiple neurodegenerative disorders. Yet not much is known about the impact of Aβ on NPC function in neurons. Here, we examined NPC and nucleoporin (NUP) distribution and nucleocytoplasmic transport using a mouse model of AD ($App^{NL-G-F/NL-G-F}$) that expresses Aβ in young animals. Our studies revealed that a time-dependent accumulation of intracellular Aβ corresponded with a reduction of NPCs and NUPs in the nuclear envelope which resulted in the degradation of the permeability barrier and inefficient segregation of nucleocytoplasmic proteins, and active transport. As a result of the NPC dysfunction *App* KI neurons become more vulnerable to inflammation-induced necroptosis – a programmed cell death pathway where the core components are activated via phosphorylation through nucleocytoplasmic shutting. Collectively, our data implicates Aβ in progressive impairment of nuclear pore function and further confirms that the protein complex is vulnerable to disruption in various neurodegenerative diseases and is a potential therapeutic target.

## Introduction

Alzheimer's disease (AD) is a progressive neurodegenerative disorder characterized by the appearance of brain pathologies such as neurofibrillary tangles and amyloid-beta (Aβ) plaques. While Aβ expression is known to disrupt different neuronal processes, the precise contribution of Aβ toward the aetiology of AD remains elusive (*De Strooper and Karran, 2016*). In the eukaryotic nuclei, nuclear pore complexes (NPCs) function as gatekeepers for the movement of macromolecules across the nuclear envelope. Each NPC is an assembly of several hundred proteins comprised of more than 30 distinct nuclear pore subunits called nucleoporins (NUPs; *Rout et al., 2000*; *Alber et al., 2007*). NUPs are further subdivided based on the spatial organization and functional domains in NPCs. The core scaffold consists of inner and outer ring NUPs, some of which are extremely long-lived proteins with low turnover rates in non-mitotic cells (*Savas et al., 2012*; *Toyama et al., 2013*). These core scaffold proteins assemble to form the central channel that contain intrinsically disordered Phe-Gly-repeat NUPs (FG-Nups). These FG-Nups have shorter half-lives and can exist in different states, including liquid-liquid phase separation, which regulates the permeability barrier in NPCs (*Celetti et al., 2020*; *Frey and Görlich, 2007*). The nuclear permeability barrier facilitates passive movement of small molecules and proteins across the central channel of the nuclear pore (<~40 kDa in molecular weight), while larger-sized proteins must be actively transported by coupling with nuclear transport proteins that recognize nuclear translocation signals (*Strambio-De-Castillia et al., 2010*). Many NUP depletion experiments that degrade the permeability barrier often disrupt nucleocytoplasmic transport of proteins and RNA (*Harel et al., 2003*; *Walther et al., 2003*; *Boehmer et al., 2003*).

Increasingly, deficits in NPC structure and function have been linked to neurodegenerative disorders (*Li and Lagier-Tourenne, 2018*). In Huntington's disease (HD), mutant huntingtin disrupts the nuclear envelope and sequesters critical nuclear transport factors such as RanGAP1 and GLE1 in the nucleus which impairs RNA export (*Gasset-Rosa et al., 2017*). In addition, NUP62 and NUP88 nuclear aggregates have also been detected in HD neurons, leading to a loss of the Ran gradient and aberrant nucleocytoplasmic transport (*Grima et al., 2017*). For amyotrophic lateral sclerosis (ALS) and frontotemporal dementia (FTD), the hexanucleotide repeat expansion in C9ORF72 is the most common cause of sporadic and familial forms of the disease. Using a genetic screen in *Drosophila*, Freibaum and colleagues identified multiple proteins in the nuclear transport machinery that modified the repeat expansion toxicity, including several scaffold NUPs that functioned as suppressors of the toxicity (*Freibaum et al., 2015*). Work done in their lab and others collectively showed that expression of the repeat expansions caused nuclear envelope abnormalities, altered composition of NPCs, mislocalized NUPs, and nucleocytoplasmic transport deficits (*Freibaum et al., 2015*; *Zhang et al., 2018*; *Coyne et al., 2020*). Interestingly, studies of TDP-43 pathology in ALS/FTD also uncovered similar links to the nuclear transport machinery, confirming that NPC dysfunction and defects in nucleocytoplasmic transport are hallmarks of the disease (*Chou et al., 2018*).

In AD neurons, abnormal localization or activation of transcriptional regulators has been reported (*Chu et al., 2007*; *Jordan-Sciutto et al., 2001*; *Ranganathan et al., 2001*; *Parra-Damas et al., 2017*; *Boissière et al., 1997*). In addition, the nuclear transport protein importin-α1 was found aggregated in Hirano bodies in AD brains (*Lee et al., 2006*). Meanwhile, in mouse AD neural stem cells, it was reported that NUP153 levels are reduced (*Leone et al., 2019*). However, none of the studies described above carried out functional experiments to test the integrity of the nucleus. In contrast to Aβ, the impact of tau on NPC has been studied in greater detail. An early imaging study in post-mortem AD brains revealed nuclear irregularities in hippocampal neurons including abnormal localization of NUP62 and NFT2 partially linked to tau pathology (*Sheffield et al., 2006*). More recently, phospho-tau is shown to interact with FG-Nups to promote aggregation and its presence disrupted NPC function and nucleocytoplasmic transport (*Eftekharzadeh et al., 2018*). In human iPSC differentiated neurons that carry an autosomal dominant mutation in tau, higher phospho-tau levels, along with abnormal invaginations in the nuclear envelope caused by microtubule invasion were observed (*Paonessa et al., 2019*). While hyperphosphorylated tau interacts with NPCs and impair nuclear function, surprisingly little is known about the impact of Aβ, if any, on nuclear integrity and NPC function.

In this paper, we examined the nuclear integrity and nucleocytoplasmic transport dynamics in an APP knock in mouse model *App^NL-G-F/NL-G-F* (*App* KI) that exhibits robust Aβ neuropathology (*Saito et al., 2014*). By using brain tissues and primary neurons cultured from *App* KI and wildtype (WT) mice, we observed a loss of NPCs in neuronal nuclei over time. Three-dimensional reconstructions of *App*

KI nuclei showed a reduction of NUP107 and NUP98 as well as an increase in the distance between individual pore complexes on the nuclear envelope in neurons at different stages of maturation. We further demonstrated that the nuclear pore dysfunction is driven by Aβ. As a consequence of this loss, the nuclear permeability barrier was compromised, leading to inefficient segregation of proteins in the nucleus and cytoplasm, and impaired active transport of NLS-tagged proteins. Finally, we designed experiments to prove that a degraded nuclear permeability barrier directly impacts neuronal viability and renders *App* KI neurons more vulnerable to TNFα-induced necroptosis, a programmed cell death pathway that requires nucleocytoplasmic shuttling for the activation of proteins found in the necrosome complex (*Weber et al., 2018*; *Yoon et al., 2016*). Taken together, our results confirms that Aβ, like other proteinopathies, can disrupt nuclear pore function in neuronal nuclei.

## Results

### *App* KI neuronal nuclei have reduced nucleoporin expression

To determine if an Aβ-expressing mouse model for AD shows altered nuclear structure and function, we studied *App* KI, a homozygous knock in mouse model that expresses mutant APP harboring three familial AD mutations expressed by the endogenous mouse promoter (*Saito et al., 2014*). While Aβ pathology is robustly detected in *App* KIs, no neurofibrillary tangles (NFT) have been reported in this mouse (*Saito et al., 2014*; *Saito et al., 2019*; *Sasaguri et al., 2017*), nor did we observe elevated hyperphosphorylated tau in *App* KI neurons (*Figure 1—figure supplement 1*). Other studies using Aβ-specific antibodies and amyloid dyes have reported the presence of diffuse and dense core plaques in *App* KI mice (*Gaunt et al., 2023*; *Clayton et al., 2021*). For this study, we established a mixed glial-neuronal hippocampal coculture from WT and *App* KI post-natal pups and allowed the cocultures to mature for up to 4 weeks (days in vitro; DIV 0-28). Cells in these coculture are healthy without axo-dendritic fragmentation or nuclear swelling, to indicate a reduction of viability throughout the duration of the experiment. The basal levels of apoptosis and necrosis in *App* KI cocultures are low, showing no difference from WT neurons (*Figure 1—figure supplement 2*). To begin, we examined the expression of NUPs in NPCs in time matched (DIV 7, 14, 21, and 28) WT and *App* KI neurons with a 'pan-NPC' monoclonal antibody (RL1) previously published to identify a group of eight NUPs from rodent nuclear envelope (*Snow et al., 1987*). We performed volumetric quantification of three-dimensionally reconstructed nuclei from confocal images and observed a reduction in the average signal intensity of NUPs in *App* KI nuclei compared to time-matched WT neurons from DIV 14-28 cocultures (*Figure 1A-B*). We also observed a similar reduction in the coverage of NUPs across the entire *App* KI nuclei (DIV 14–28; *Figure 1A and C*). To complement the coculture studies, we assessed NUP intensities from pyramidal neurons in the anterior *cornu ammonis* 1 (CA1) hippocampal region of WT and *App* KI mice at 4-, 7-, and 13 months old. Just like in cocultured neurons, we saw a loss of NUP signal in individual *App* KI neuronal nuclei in mice as early as 4 months (*Figure 1D–E*). Of note, we saw a reduction of NUP signal in 13-mo. old WT animals as compared to younger animals, indicating an age-dependent decline of the signal over time.

To further resolve the differences in distribution of NUPs, we used Zeiss AiryScan to capture and enhance the spatial resolution of the fluorescent signals on the nuclear envelope. Previous studies using RL1 antibodies have shown distinct fluorescent puncta that colocalized with different NUPs (*Guan et al., 2000*; *Cheng et al., 2019*). These labeled structures were identified as NPCs via immunoelectron microscopy (*Snow et al., 1987*). We confirmed these studies with high-resolution confocal imaging by demonstrating that the majority of RL1 fluorescent puncta on the nuclear envelope colocalized strongly with NUP98 (Pearson's correlation coefficient, $r=0.9541$ *Figure 1—figure supplement 3*). We then used IMARIS to identify and quantify RL1 puncta and saw that the average number of puncta per nuclei was reduced in *App* KI neurons across all time points examined, including a subtle decrease in DIV 7 neurons which was previously undetected (*Figure 1F–G*). We next performed nearest neighbor calculations for each NPC puncta and showed that *App* KI neurons also had higher average distances between puncta for all nearest neighbor (1, 3, 5, and 9) calculations, suggesting that NPCs are more sparsely distributed in *App* KI relative to WT neurons (*Figure 1F, H-I*). Notably, the largest reduction in NPC counts and distribution in *App* KI neurons occurred when neurons matured from DIV 7-14 (*Figure 1G–I*). Finally, all the *App* KI NPC deficits described above were replicated independently using another NPC antibody (mAb414; *Figure 1—figure supplement 4*).

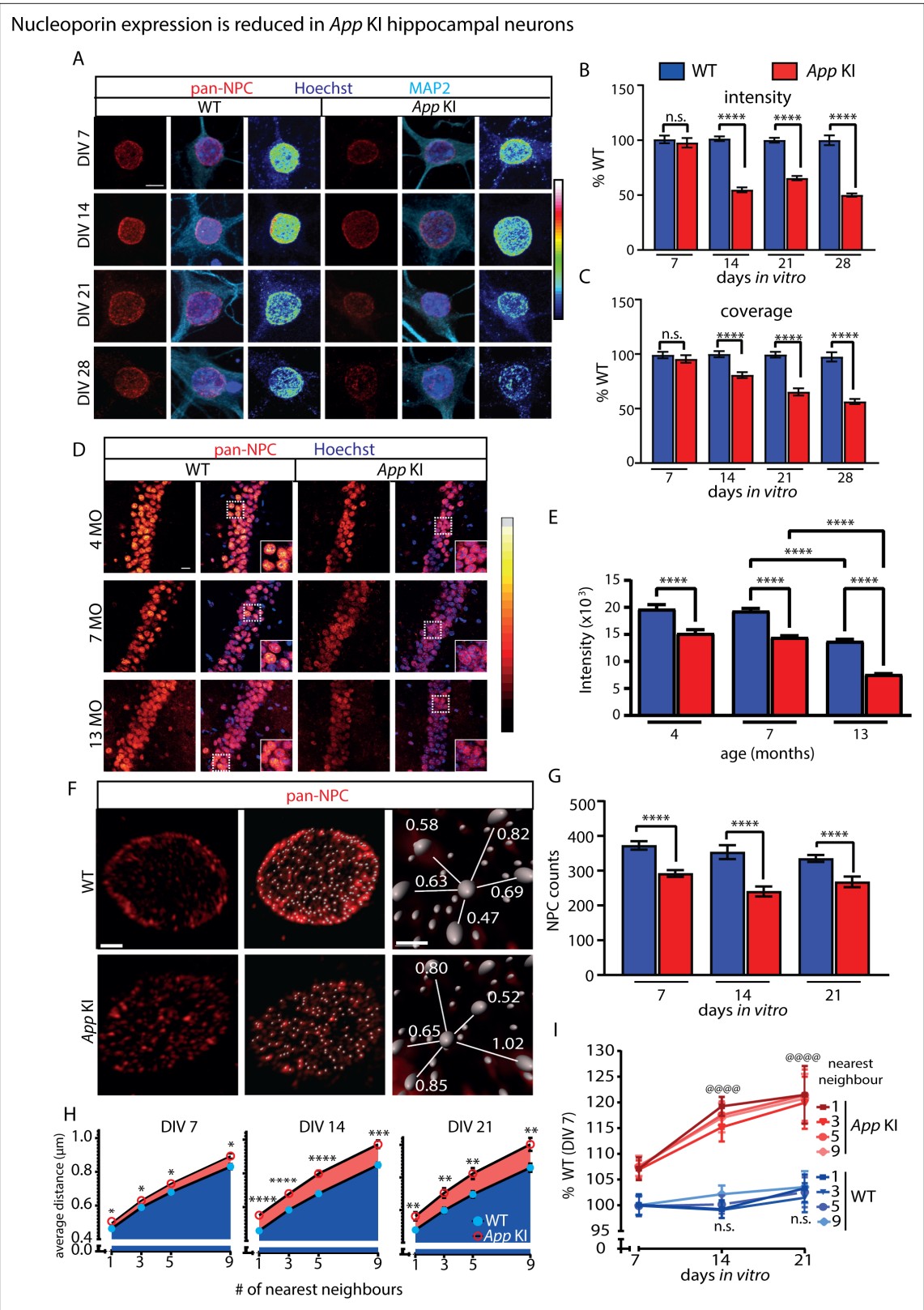

**Nucleoporin expression is reduced in *App* KI hippocampal neurons**

**Figure 1.** NUP intensity and coverage are decreased in *App* KI neurons. (**A**) Representative confocal images nuclei of WT and *App* KI hippocampal neurons (DIV 7–28). Cells were co-labeled with antibodies against pan-NPCs (red), MAP2 (cyan), and Hoechst (blue). A heatmap for pan-NPC expression is included. Scale bar at 5 µm. (**B-C**) Volumetric quantification of intensity and coverage of NPCs in WT and *App* KI nuclei (DIV 7–28). Bar graph is plotted as percent time-matched WT neurons for each condition across independent experiments (n=45 neurons/group across three replicates for DIV 7, 21,

*Figure 1 continued on next page*

Figure 1 continued

and 28; n=90 neurons/group across six replicates for DIV 14). (**D**) Representative confocal images of brain sections in hippocampal CA1 pyramidal cell layer from WT and *App* KI mice at different ages (4, 7, and 13 months). Slices were labeled with antibodies against NPCs (red), Hoechst (blue). Scale bar at 20 µm. Inset image is a magnification of the area outlined by a white dotted box. (**E**) Quantification of pan-NPC intensity from individual neuronal nuclei in (**D**) as outlined in methods. Group data plotted as percent change from average age-matched WT (4, 7, and 13 months.; n=120 neurons/group across three replicates). (**F**) Zeiss AiryScan confocal images of hippocampal neurons (DIV 14) labeled with pan-NPCs antibodies (red) after linear deconvolution. Representative spot masks and nearest neighbor distances shown. Scale bars are 1 µm (left) and 0.5 µm (right). (**G**) Quantification of total NPC puncta per nuclei from WT and *App* KI neurons from time-matched neuronal cultures (n=14 nuclei/group averaged across two replicates). (**H-I**) Quantification of average distance between nearest neighbors (1, 3, 5, and 9 nearest neighbors) for individual NPC puncta. The average distance for each nearest neighbor is plotted as individual line graphs for WT and *App* KI neurons across time-matched cultures (DIV 7–21) and also in (**I**) where it is normalized against the average signal from DIV 7 WT neurons (n=14 nuclei/group across two replicates). Significance values in (**I**) represent change from DIV 7 for each genotype. For all graphs values in blue represent WT and values in red represent *App* KI. Unless otherwise stated, all data are represented as mean ± SEM. Depending on distribution of data, significance testing between WT and *App* KI samples were done using Mann-Whitney U test (**B, E**), unpaired t-test (**C, G, H,**) or two-way ANOVA (**I**). For all statistics significance is as follows: not significant (ns), <0.05 (*), <0.001 (**), <0.0001 (***), <0.0001 (****/@@@@). For complete statistical output, refer to **Supplementary file 1-figure 1**.

The online version of this article includes the following figure supplement(s) for figure 1:

**Figure supplement 1.** WT and *App* KI neurons have comparable levels of hyperphosphorylated Tau.

**Figure supplement 2.** *App* KI and WT cocultures show similarly low levels of apoptosis and necrosis at basal state.

**Figure supplement 3.** pan-NPC (RL1) antibodies strongly colocalizes with NUP98.

**Figure supplement 4.** pan-NPC (mAB414) recapitulates NPC deficits in *App* KI neuronal nuclei.

We next examined if select NUP expression in NPCs are differentially expressed by looking at two NUPs: NUP107, a long-lived NUP that forms part of the NPC scaffold, and NUP98, a mobile FG-repeating NUP located at both the nuclear and cytoplasmic sides of the central core (**Walther et al., 2003**; **Griffis et al., 2003**). In *App* KI cocultures, both NUP98 and NUP107 had lower levels of expression relative to WT neurons (DIV 14–21; **Figure 2A–D**). While the loss of NUP98 is sustained from DIV7 to DIV21, the reduction of NUP107 levels occurs in DIV14 neurons and is sustained in DIV21 neurons. High-resolution confocal imaging of WT and *App* KI cocultures confirmed a reduction of NUP98-positive puncta similar to those observed using the pan-NPC RL1 antibodies, thus supporting the idea that loss of selected NUPs translates to lower NPC counts and sparser distribution on the nuclear envelope (**Figure 2—figure supplement 1**). To complement cocultures experiments, IHC of brain sections from WT and *App* KI animals also showed a decline in NUP98 and NUP107 levels in CA1 pyramidal neurons across all ages tested (**Figure 2E–G**). Interestingly, neuronal nuclei from younger *App* KI animals at 2 months already exhibit a reduction in NUP98 and NUP107 despite low numbers of Aβ plaques at this age (Figure 5A; **Saito et al., 2014**). To further confirm that the altered NUP levels can also be detected in other Aβ-expressing mouse models, we performed IHC on forebrain sections of 4–5 months old APP/PS1 (APPSwe/PSEN1-A246E) mice and saw a similar reduction in NUP98 and NUP107 levels in CA1 pyramidal neurons (**Figure 2—figure supplement 2**). Finally, we isolated nuclei from 2- and 14 months old WT and *App* KI mouse forebrains via density gradient centrifugation for immunoblotting to confirm the loss of NUP98 and NUP107 (**Figure 2H–I**). We noted that protein levels and subcellular localization of LAMIN-B1, and the morphology of the nuclear envelope as defined by the sphericity index remained comparable between WT and *App* KI neurons, suggesting that the overall architecture of the nucleus is unperturbed in the mutant (**Figure 2H**; **Figure 2—figure supplement 3**). Taken together, our experiments collectively demonstrate a robust reduction of selected NUPs in *App* KI neuronal nuclei, even at early stages of the disease.

## Expression of Aβ correlates with a loss of NUPs in neurons

It remains unclear if Aβ expression directly contributes to a reduction of NUPs and how that occurs. One possibility is that increased levels of intracellular Aβ mediates the loss of NUPs. We quantified the presence of intracellular Aβ in a majority of our experiments using two different antibodies (MOAB-2 and 82E1) raised against Aβ (**Figure 3A–B**). In *App* KI mice, a diffuse intracellular Aβ can be detected in animals as early as 2 months. In older mice, there is an increase of punctate Aβ-positive aggregates in hippocampal neurons (**Figure 3—figure supplement 1**). Similarly, in neuron cocultures, there was an increase in intracellular Aβ levels over WT neurons that parallels the reduction of NUPs as neurons mature from DIV 7–28 (**Figure 3A–C**; **Figure 3—figure supplement 2**).

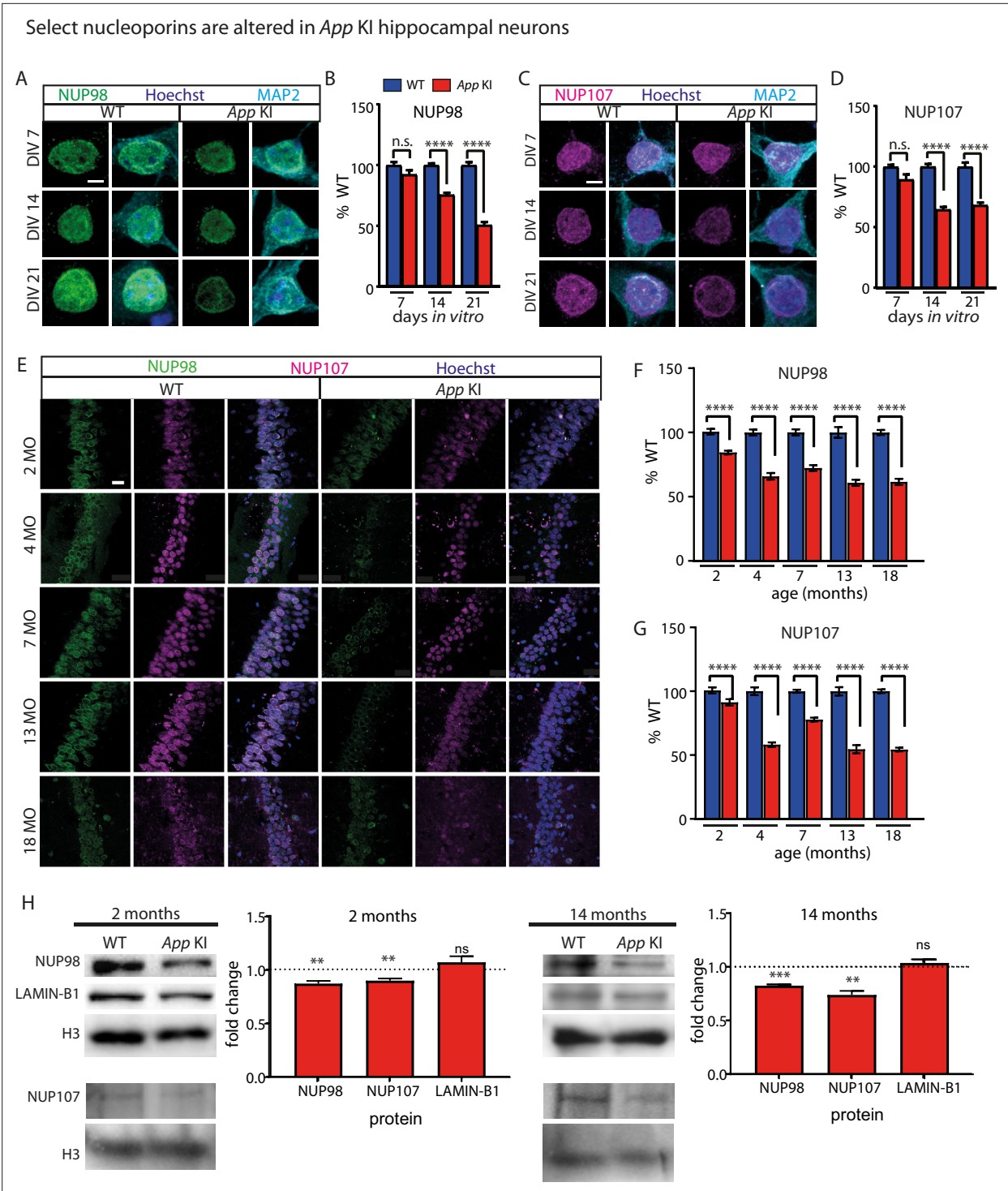

**Figure 2.** NUP98 and NUP107 are reduced in *App* KI neurons. (**A**) Representative confocal images for WT and *App* KI neurons (DIV 7–21) immunolabeled with antibodies against NUP98 (green), MAP2 (cyan), and Hoechst (blue). Scale bar at 5 μm. (**B**) Volumetric quantification of NUP98 intensity in WT and *App* KI nuclei (DIV 7–21). Bar graph is plotted as percent time-matched WT neurons for each condition across independent experiments (n=45 neurons/group across three replicates for DIV 7, and 21; n=75 neurons/group across five replicates for DIV 14). (**C**) Same as in (**A**) except neurons are labeled with NUP107 (Magenta). Scale bar at 5 μm. (**D**) Same as in (**B**) except quantifications are done for NUP107 (n=45 neurons/group across three replicates for DIV 7, and 21; n=75 neurons/group across five replicates for DIV 14) (**E**) IHC of brain slices from *App* KI (2, 4, 7, 13, and 18 months) and age-matched WT mice. Images show hippocampal CA1 pyramidal layer neurons. Sections were labeled with antibodies against NUP98 (green), NUP107 (magenta), and Hoechst (blue). Scale bar at 10 μm. (F-G) Quantification of NUP98 and NUP107 intensities from individual neurons shown in (**E**). Imaging parameters and methodology of cell selection are outlined in methods. Group data plotted as percent age-matched WT for each

*Figure 2 continued on next page*

*Figure 2 continued*

age group (2, 4, 13, and 18 months- n=90 neurons/group across three animals; 7 months- n=210 neurons/group across seven animals). (**H**) Western blots of purified forebrain nuclear extracts from 2- and 14-month-old WT and *App* KI animals. Blots were immunoassayed for antibodies against NUP98, NUP107, LAMIN-B1, and Histone H3. For fold change quantification, all bands were normalized against H3 and analysed with one sample t-test (N=5 for 2 months or N=4 for 14 months) paired experiments from WT and *App* KI animals. Dotted line on Y axis indicate a fold change of 1.0. For all graphs values in blue represent WT and values in red represent *App* KI. For all graphs, data was reported as mean ± SEM. Depending on distribution of data, significance testing were performed using Mann-Whitney U test (B, D, F, G) or one sample t-test (2 H). For all statistical significance is as follows: not significant (ns), <0.05 (*), <0.01 (**), <0.001 (***), <0.0001 (****). For complete statistical analysis, refer to *Supplementary file 1*.

The online version of this article includes the following source data and figure supplement(s) for figure 2:

**Source data 1.** PDF file containing original western blots for *Figure 2H* indicating the relevant bands and treatments.

**Source data 2.** Individual raw western blots for *Figure 2H* indicating bands and treatments.

**Figure supplement 1.** Reduction of NUP98 puncta in *App* KI nuclei.

**Figure supplement 2.** Reduced NUP98 and NUP107 expression in APP/PS1 animals.

**Figure supplement 3.** LAMIN-B1 expression and nuclear sphericity is comparable in *App* KI and WT neurons.

To assess whether cleavage of Aβ compromises the nuclear integrity of *App* KI neurons, we blocked γ-secretase activity with DAPT to inhibit the cleavage of mutant APP in cocultures. DAPT reduces Aβ in primary neuronal cultures and in brain extracts (*Dovey et al., 2001*; *Lanz et al., 2003*; *Kienlen-Campard et al., 2002*). Prior to treatment with DAPT, we washed and replenished 50% of the conditioned media to remove pre-existing extracellular Aβ present in the media (*Figure 3D*). We reasoned that at DIV7, the extracellular media is unlikely to be saturated with Aβ and removal of half the conditioned media per day for 3 consecutive days should not compromise neuronal viability. We then exposed DIV 10 cocultures to DAPT at 24 hr intervals for 4 days to ensure constant blockade of γ-secretase activity throughout the duration of the experiment. As controls, we performed matching treatments in WT neurons to show that DAPT treatment did not alter neuronal viability. Our results using pan-NPC RL1 antibody labeling showed that treatment of *App* KI neurons with DAPT partially restored both the coverage and average intensity of NUPs in the nuclei relative to DMSO controls, suggesting that inhibition of γ-secretase activity halted intracellular Aβ accumulation and prevented further deterioration of the nucleus and loss of NUPs (*Figure 3E–G*). While DAPT did not completely restore expression to WT levels, the partial loss is likely attributed to pre-existing conditions in DIV7 *App* KI neurons, or the continuous impact of the residual Aβ present in the remaining conditioned media at the start of the experiment. Importantly, we noted that DAPT treatment also reduced the amount of intracellular Aβ (*Figure 3H*) which inversely correlates with the loss of NUP expression.

We next asked if direct addition of synthetic Aβ42 in WT cocultures can recapitulate the nuclear dysfunction in *App* KI neurons. We exposed WT (DIV 10) cocultures to either monomeric, oligomeric or fibrillar Aβ42 for 4 days before performing ICC on these cocultures at DIV 14 (*Figure 3I–J*). Remarkably, only neurons exposed to oligomeric but not monomeric or fibrillar Aβ42 showed a reduction in intensity and coverage of NUPs (*Figure 3K–L*) and a corresponding increase in intracellular Aβ (*Figure 3M*). No Aβ42 was detected in neurons incubated with monomeric Aβ42, while larger clumps of fibrillar Aβ42 were detected only in the extracellular space (*Figure 3—figure supplement 3*). In agreement with our hypothesis, the accumulation of oligomeric Aβ42 in neuronal soma correlated with the loss of NUPs, thus directly linking the presence of the toxic form of Aβ with NPC dysfunction.

Finally, we asked if extracellular Aβ released into the media from *App* KI neurons is sufficient to alter NUP levels in WT neurons. To that end, we performed a media exchange experiment between WT and *App* KI neurons where the harvested conditioned media was reciprocally plated between these two cocultures (from DIV19 to 21; *Figure 3N*). We observed that in the presence of *App* KI condition media (WT +*appm*), WT neurons showed a robust loss of intensity and coverage of NUPs similar to that observed in the *App* KI control group (*App* KI +*appm*), implying that internalization of extracellular Aβ from the media is sufficient to reduce NPC levels (*Figure 3O–Q*). Indeed, for these WT neurons incubated with *App* KI media, we recorded an increase in intracellular Aβ which is presumably up taken from the extracellular environment (*Figure 3R*). We also noted that the incubation of WT conditioned media in *App* KI cocultures (*App* KI +*wtm*) failed to reverse the loss of NUPs, as it is likely that any reduction of NUPs prior to media swap is either irreversible or cannot be restored within the

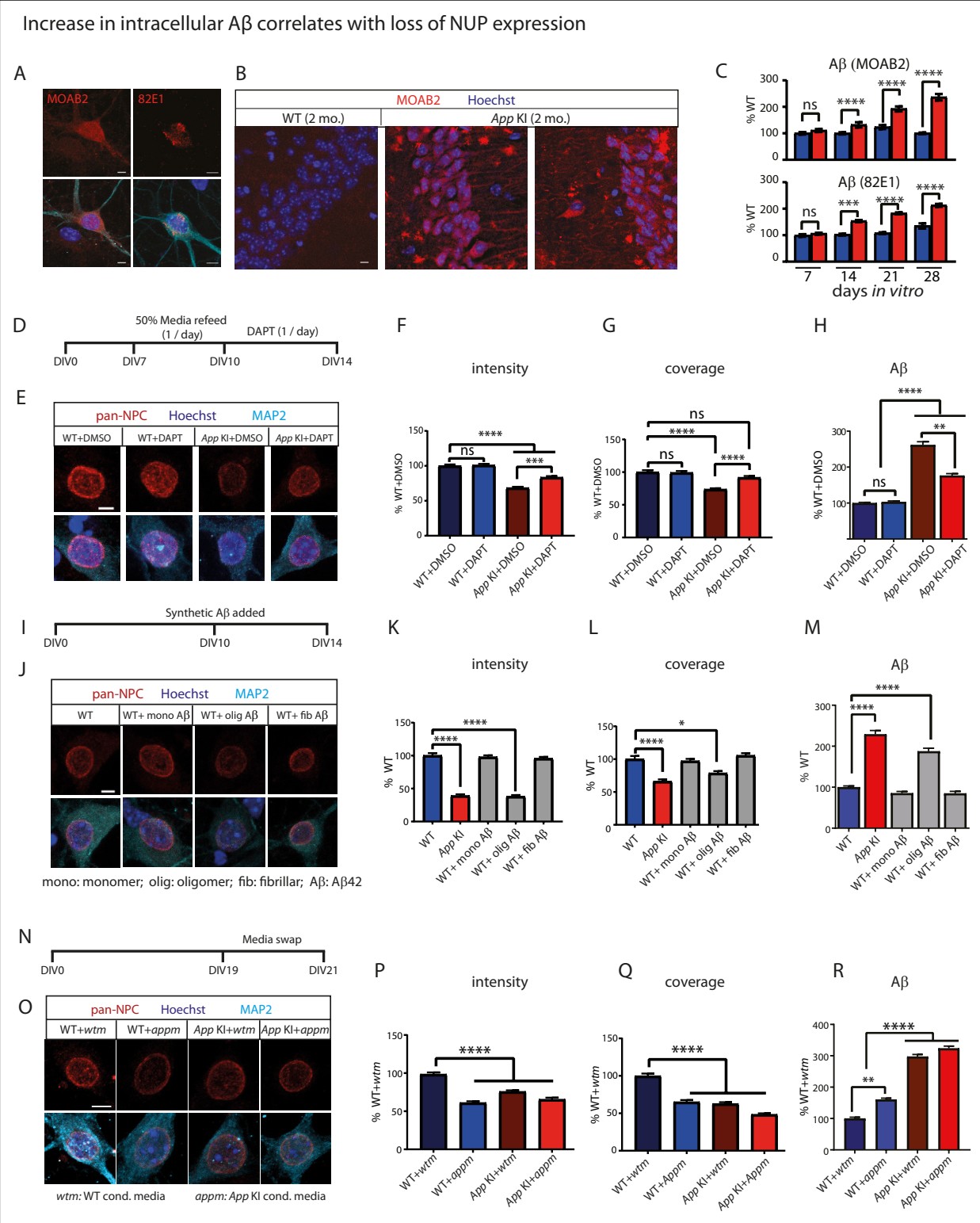

Figure 3. Increase in intracellular Aβ correlates with loss of NUPs. (**A**) Representative ICC images of DIV 14 *App* KI cultured neurons stained with two Aβ antibodies MOAB-2 (red) and 82E1 (red), MAP2 (cyan) and Hoechst (blue) showing intracellular Aβ accumulations. Scale bar at 5 µm. (**B**) IHC confocal image of CA1 pyramidal cell region of WT and *App* KI 2 months mice stained with MOAB-2 (red) and Hoechst (blue). Scale bar at 20 µm. (**C**) Quantification of Aβ with MOAB-2 and 82E1 antibodies in DIV 7–28 cocultured neurons. Values shown as percent change from time-matched WTs n=45 cells/group across three replicates for MOAB-2 (DIV 7, 21, and 28) for 82E1 (DIV 7–28); n=90 cells/group across six replicates for MOAB-2 (DIV 14). (**D**) Experimental timeline for inhibition of APP cleavage using DAPT. Media from cocultures were partially replaced from DIV 7–10 to remove

*Figure 3 continued on next page*

*Figure 3 continued*

pre-existing Aβ, then DAPT was added (10 µM) every 24 hr. from DIV 10–14. (**E**) ICC of WT and *App* KI neurons incubated with DMSO (control) or DAPT stained for NPCs (red), MAP2 (cyan), and Hoechst (dark blue). Scale bar at 5 µm. (**F-H**) Quantification of intensity (**F**), coverage (**G**) of NPCs and intracellular Aβ (MOAB-2; **H**) in WT and *App* KI neurons exposed to DMSO and DAPT. Values shown as a percent change from time-matched WT neurons exposed to DMSO (n=45 cells/group across three replicates). (**I**) Experimental timeline for addition of exogenous Aβ42 addition to WT cocultures. Different preparations of synthetic Aβ42 was added to cultures from DIV 10–14. (**J**) ICC images of WT neurons exposed to monomeric, oligomeric and fibrillar Aβ42 stained for NPCs (red), MAP2 (cyan), and Hoechst (dark blue). Scale bar at 5 µm. (**K-M**) Quantification of intensity (**K**) and coverage (**L**) of NPCs and intracellular Aβ (MOAB-2; **M**) in WT and *App* KI cocultures exposed to synthetic Aβ42. Values shown as a percent change from WT (n=45 cells/group across three replicates). (**N**) Timeline for media swap experiment. Conditioned media derived from WT and *App* KI cocultures were swapped and incubated for 2 days (DIV 19–21) before ICC. (**O**) Confocal images of WT and *App* KI neurons exposed to WT conditioned media (*wtm*) and *App* KI conditioned media (*appm*). Cells were co-stained with pan-NPCs (red), MAP2 (cyan), and Hoechst (dark blue). Scale bar at 5 µm. (P-R) Quantification of intensity (**P**) and coverage (**Q**) of NPCs and intracellular Aβ intensity (MOAB-2; **R**) in WT and *App* KI cultures exposed to *wtm* or *appm*. Values shown as a percent change from WT culture exposed to WT media (WT +*wtm*; n=45 cells/group across three replicates). For all graphs values in blue represent WT and values in red represent *App* KI. All data are reported as mean ± SEM. Unless otherwise stated, all significance tests were conducted with Mann-Whitney U test (**C**) or one-way ANOVA Kruskal Wallis; F, H, K, L, M, P, Q, R. Tukey; (**G**). For all statistics significance is as follows: not significant (ns), <0.05 (*), <0.01 (**), <0.001 (***), <0.0001 (****). For complete statistical profiles for each experiment, refer to ***Supplementary file 1***.

The online version of this article includes the following figure supplement(s) for figure 3:

**Figure supplement 1.** Immunohistochemistry of *App* KI mice show Aβ plaques and punctate aggregates.

**Figure supplement 2.** Localization of Aβ in *App* KI neurons.

**Figure supplement 3.** Subcellular accumulation of exogenously added synthetic Aβ42 in WT neurons.

time frame of the experiment. Collectively, these Aβ coculture experiments strongly indicates that expression of Aβ is sufficient to trigger a reduction in NUP expression in neurons.

## Compartmentalization of nucleocytoplasmic proteins is impaired in *App* KI neurons

A loss of NUP expression or changes in NPC distribution can compromise the nuclear permeability barrier and weaken nucleocytoplasmic compartmentalization of proteins and macromolecules. We predicted that a degraded size-exclusion barrier will permit an increase in unregulated movement of segregated proteins between the nucleus and cytoplasm. To test these ideas, we performed a series of fluorescence recovery after photobleaching (FRAP) experiments using enhanced green fluorescent protein (EGFP) constructs transiently expressed in neuronal cocultures (DIV 14) from WT and *App* KI pups. We first examined the nucleocytoplasmic movement of monomeric EGFP, a relatively low molecular weight (MW) protein (~27 kDa) that can passively diffuse across the NPC and does not require active transport. We partially photobleached EGFP in the nucleus and tracked the recovery of fluorescence in the same region (***Figure 4A–B***; magenta dotted circle). In both groups of neurons, we observed a rapid recovery of GFP fluorescence after photobleaching, with the signal reaching a plateau after 5 min of recovery. However, relative to WT neurons, *App* KI neurons had a faster rate of recovery, suggesting that the cytoplasmic pool of unquenched EGFP and other similarly sized proteins can enter the nucleus more readily (***Figure 4A–D***).

Since passive movement of small proteins across the nuclear pore is altered, we reasoned that movement of larger MW proteins into the nucleus might be similarly impacted. We transiently expressed a 4xEGFP construct which expresses a single fusion protein with four tandem copies of EGFP (~108 kDa) in hippocampal neurons. In the absence of any nuclear translocation signals (NTS), this larger MW protein is normally sequestered in the cytoplasm and mostly excluded from the nucleus. However, prior to time-lapse imaging experiments, we already observed a moderate increase in nuclear-localized 4xEGFP in transfected *App* KI under steady state conditions (***Figure 4G***; ***Figure 4—figure supplement 1***). Thus, we photobleached the nucleus prior to FRAP to eliminate any pre-existing fluorescence before quantifying the recovery of nuclear GFP via time-lapse imaging (***Figure 4E***; magenta dotted circle). Results show that in WT neurons, there was no change in GFP signal throughout the duration of the time-lapse imaging experiment (~1 h), indicating that the 4xEGFP was unable to enter the nucleus. However, unlike WT neurons, GFP fluorescence steadily increased in *App* KI neurons over the duration of the experiment, suggesting that the compromised permeability barrier allowed nuclear entry of high MW proteins normally sequestered in the cytoplasm (***Figure 4E–H***).

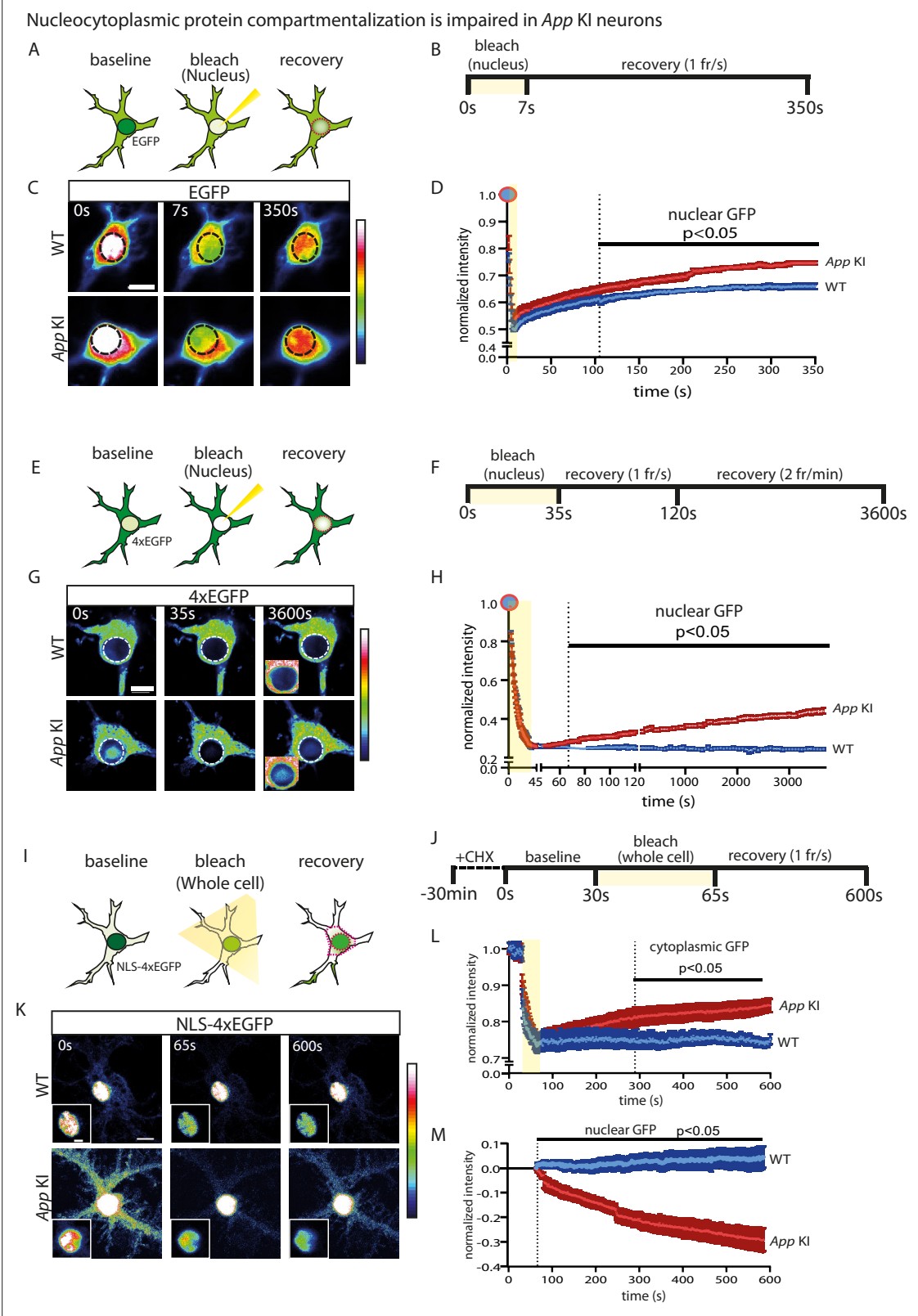

Nucleocytoplasmic protein compartmentalization is impaired in *App* KI neurons

**Figure 4.** Nucleocytoplasmic protein compartmentalization is impaired in *App* KI neurons. (**A**) Schematic of FRAP experiment in transfected neurons expressing EGFP. EGFP is photobleached in the nucleus and recovery of fluorescence is tracked in the nucleus (magenta dotted line). (**B**) Time course for EGFP FRAP experiment and imaging parameters. (**C**) Representative static images with time stamp (t=0, 7, and 350 s) of WT and *App* KI neurons captured during FRAP. Black dotted line represents the photobleached ROI. Scale bar at 10 μm. (**D**) FRAP recovery curve for EGFP in the nucleus. All

*Figure 4 continued on next page*

*Figure 4 continued*

data points are normalized against baseline values prior to photobleaching. Shaded region (yellow) signifies photobleaching period. Dotted vertical line indicates all time points post t=105 s that are p<0.05 between WT and *App* KI (n=8 neurons/group across two replicates). (**E**) Schematic of FRAP experiment in transfected neurons expressing 4xEGFP. 4xEGFP is photobleached and fluorescence recovery is tracked in the nucleus (magenta dotted line). (**F**) Time course for 4xEGFP FRAP experiment and imaging parameters. Image capture was performed at two frame rates: 1 frame/s (t=35–120 s) followed by 2 frames/min (t=121–3600 s). (**G**) Representative static images with time stamps (t=0, 35, and 3600 s) of WT and *App* KI neurons captured during FRAP. White dotted line represents the photobleached region. Scale bar at 10 µm. Inset image showing over-saturated signal to highlight 4xEGFP nuclear accumulation. Scale bar at 10 µm. (**H**) FRAP recovery curve for 4xEGFP in the nucleus. All data points are normalized against baseline values prior to photobleaching. Shaded region (yellow) signifies photobleaching period. Dotted vertical line indicates all time points post t=67 s that are p<0.05 between WT and *App* KI (n=10 neurons/group across three replicates). (**I**) Schematic of FRAP experiment for transfected neurons expressing NLS-4xEGFP. Whole cell photobleaching was performed and fluorescence recovery tracked in the nucleus and cytoplasm (magenta dotted lines). (**J**) Time course for NLS-4xEGFP FRAP experiment and imaging parameters. (**K**) Representative static images with time stamps (t=0, 65, and 600 s) of WT and *App* KI neurons captured during FRAP. Images are intentionally oversaturated (for the nucleus) to show differences in cytoplasmic NLS-4xEGFP. Scale bar at 10 µm. Inset image shows NLS-4xEGFP signal in the nucleus at normal saturation levels to capture differences in the nucleus. Scale bar is 5 µm. (**L**) Recovery curve for NLS-4xEGFP in the cytoplasm. All data points are normalized against baseline values prior to photobleaching. Shaded region (yellow) indicates photobleaching period. Dotted vertical line indicates all time points post t=289 s that are p<0.05 between WT and *App* KI (n=10 neurons/group across three replicates). (**M**) Quantification of nuclear NLS-4xEGFP post-photobleaching. All values are normalized against data point at t=65 s (first time point post-photobleaching). Dotted vertical line indicates all time points after t=70 s that are p<0.05 between WT and *App* KI (n=10 neurons over three replicates). For all graphs, values in blue represent WT and values in red represent *App* KI. All graphs are reported as mean values ± SEM. For all recovery curves, significance tests performed with Mann-Whitney U test. For complete statistical profiles for each experiment, refer to *Supplementary file 1*.

The online version of this article includes the following figure supplement(s) for figure 4:

**Figure supplement 1.** *App* KI neurons have enhanced accumulation of cytoplasmic proteins in the nucleus.

**Figure supplement 2.** *App* KI neurons have increased cytoplasmic accumulation of NLS-tagged proteins.

**Figure supplement 3.** Enhanced accumulation of CRTC1 in *App* KI nuclei in basal and TTX-silenced states.

If cytoplasmic proteins are aberrantly entering the nucleus in *App* KI neurons, then a reverse flow of proteins from nucleus-to-cytoplasm may exist to counterbalance the continuous influx of proteins into the nucleus. Such a mechanism would prevent sustained accumulation of proteins which would quickly overwhelm the nucleus and become toxic. To examine if loss of NUPs resulted in bidirectional deficits in nucleocytoplasmic movement of proteins, we transfected hippocampal neurons with an SV40 nuclear localization signal (NLS) tagged 4xEGFP, a fusion protein that is actively transported into the nucleus via the classical nuclear import pathway. As expected, expression of NLS-4xEGFP in both WT and *App* KI neurons showed a strong concentration of GFP fluorescence in the nucleus (*Figure 4K*). Prior to FRAP, we imaged fixed neurons and observed elevated cytoplasmic NLS-4xEGFP signal in the transfected *App* KI neurons under steady state conditions, likely signaling a disrupted permeability barrier (*Figure 4K*, *Figure 4—figure supplement 2*). However, the increase in cytoplasmic GFP could still be attributed to other changes in the cytoplasmic milieu such as enhanced protein synthesis. To eliminate that possibility, we pre-incubated cocultures with cycloheximide (CHX) to halt protein synthesis prior to live cell imaging. In order to study the nucleocytoplasmic movement in isolation, we photobleached the GFP fluorescence across the entire neuron to eliminate pre-existing fluorescence, targeting as many distal dendrites as possible to rule out retrograde movement of unquenched GFP from neuronal processes (*Figure 4I*). While we could not avoid partial photobleaching of the nuclear GFP signal, the loss of GFP intensity is comparatively mild given the overwhelming accumulation of NLS-4xGFP in the nucleus (*Figure 4K*). After photobleaching, we tracked and quantified changes in GFP fluorescence in the cytoplasm and nucleus over the duration of the experiment (*Figure 4I*; magenta dotted regions). In WT neurons, we saw no change in either cytoplasmic or nuclear GFP fluorescence which remained largely confined to the nucleus after photobleaching (*Figure 4K–M*). In contrast, there was a marked increase in cytoplasmic GFP with a corresponding decrease in nuclear GFP fluorescence after photobleaching of *App* KI neurons, indicating a leaky nucleus that promotes aberrant nucleus-to-cytoplasmic movement of NLS-4xEGFP (*Figure 4K–M*).

Finally, we examined the localization of an endogenous nucleocytoplasmic shuttling protein CRTC1, a transcriptional coactivator that responds to neural activity by translocating from the synapse and cytosol to the nucleus (*Lim et al., 2017*; *Ch'ng et al., 2012*; *Ch'ng et al., 2015*). Under basal condition, there is elevated CRTC1 in *App* KI nuclei compared to WT neurons, an observation that aligns with the compromised protein compartmentalization in *App* KI neurons (*Figure 4—figure supplement 3*).

To rule out any differences in spontaneous activity that may impact CRTC1 nuclear localization, we inhibited action potentials by incubating time-matched WT and *App* KI cocultures with tetrodotoxin (TTX) and the results were similar (*Figure 4—figure supplement 3*). Overall, our experiments indicate a compromised nuclear permeability barrier in *App* KI neurons that leads to an unregulated, bidirectional movement of nuclear and cytoplasmic proteins across the nuclear membrane.

## The nuclear permeability barrier is compromised in *App* KI animals

A decline in NPC function and disruption in nucleocytoplasmic transport occurs during the normal aging process (*Mertens et al., 2015*; *D'Angelo et al., 2009*). We hypothesized that the loss of NUP expression in *App* KI animals would accelerate the degradation of the size-exclusion permeability barrier. Given that the loss was detected early in coculture neurons (DIV 7) and in *App* KI animals (2 months), we asked if the permeability barrier is also disrupted during early stages of the disease by comparing nuclei isolated from 2- and 14-month-old WT and *App* KI animals. At 2 months, IHC of *App* KI forebrain sections showed sparsely distributed Aβ plaques across the cortex and hippocampus. In contrast, by 13 months, larger-sized plaques are present throughout the entire brain, including all the cortical layers and the hippocampus (*Figure 5A*; *Saito et al., 2014*). We extracted and incubated nuclei from these animals and added a mixture of fluorescently conjugated dextrans that comprised of low (70 kDa-FITC-dextran) and high (500 kDa rhodamine-dextran) MW variants. We anticipated that high MW dextrans would be excluded from unruptured nuclei while low MW dextrans would differentially enter the nuclei depending on the state of the permeability barrier. Indeed, our results indicated that *App* KI nuclei are leaky, with a higher proportion of low MW dextrans found in the *App* KI nuclei for both 2- and 14 months old animals, and with older animals showing a much stronger deficit (*Figure 5B–C*). Reassuringly, control experiments indicated that the nuclei did not rupture during the extraction process as the high MW dextrans were equally excluded from WT and *App* KI nuclear isolates (*Figure 5D*). In short, we conclude that the nuclear permeability barrier is compromised in *App* KI animals as well as in vitro and the damage can be detected at initial stages of the disease in mice, prior to robust presence of Aβ plaques.

## Active transport of NLS-tagged proteins is impaired in *App* KI neurons

To test if proteins entering the nucleus via the classical nuclear import pathway are also disrupted, we again employed FRAP to track active import of NLS-4xEGFP into the nucleus by photobleaching and quantifying the recovery of GFP fluorescence in the nucleus. Since active transport of NLS-tagged proteins is an energy-dependent process that occurs with faster kinetics, we shortened our photobleaching duration of nuclear NLS-4xEGFP and tracked fluorescence recovery for up to 45 min (*Figure 6A–B*; magenta dotted circle). For WT nuclei, GFP fluorescence recovered rapidly in the first 30 s, reaching a plateau within 60 s post-photobleaching (*Figure 6C–D*). In contrast, fluorescence recovery was delayed in *App* KI nuclei, with a more gradual increase of GFP fluorescence over time, eventually matching WT levels of recovery after ~30 min. post-bleaching. An analysis of the slope of recovery indicates that for the first 30 s, GFP fluorescence recovered approximately three times faster in WT compared to *App* KI neurons. One possibility for the disruption of nucleocytoplasmic transport is that loss of NUPs destabilizes NPCs and reduces docking of the importin-α/β1 heterodimeric nuclear adaptor complex at the nuclear pore. ICC experiments on time-matched cocultured neurons showed a distinct reduction in the average intensity and coverage of importin β1 (IMP-β1) signal around the nucleus in *App* KI neurons that is dependent on maturity of the cocultures (*Figure 6E–G*). Apart from IMP-β1, the intensity of RanGAP1 was also significantly lower in *App* KI neurons (*Figure 6H–I*). RanGAP1 is tethered to the NPC and is critical for Ran GTPase-mediated active transport of nuclear proteins (*Matunis et al., 1998*). It is highly likely that a loss of both IMP-β1 and RanGAP1 will negatively impact the nuclear transport of proteins. Collectively, our data argues that loss of NUPs in *App* KI neurons also leads to a disruption of the nuclear import of NLS-proteins via the classical importin α/β1-mediated pathway.

## Nuclear pore complex dysfunction increases neuronal vulnerability toward TNF-α induced necroptosis

The cellular mechanisms for AD-associated neuronal loss remains unclear. Recent reports in postmortem AD brains and in animal models for AD have linked necroptosis as a potential cell death

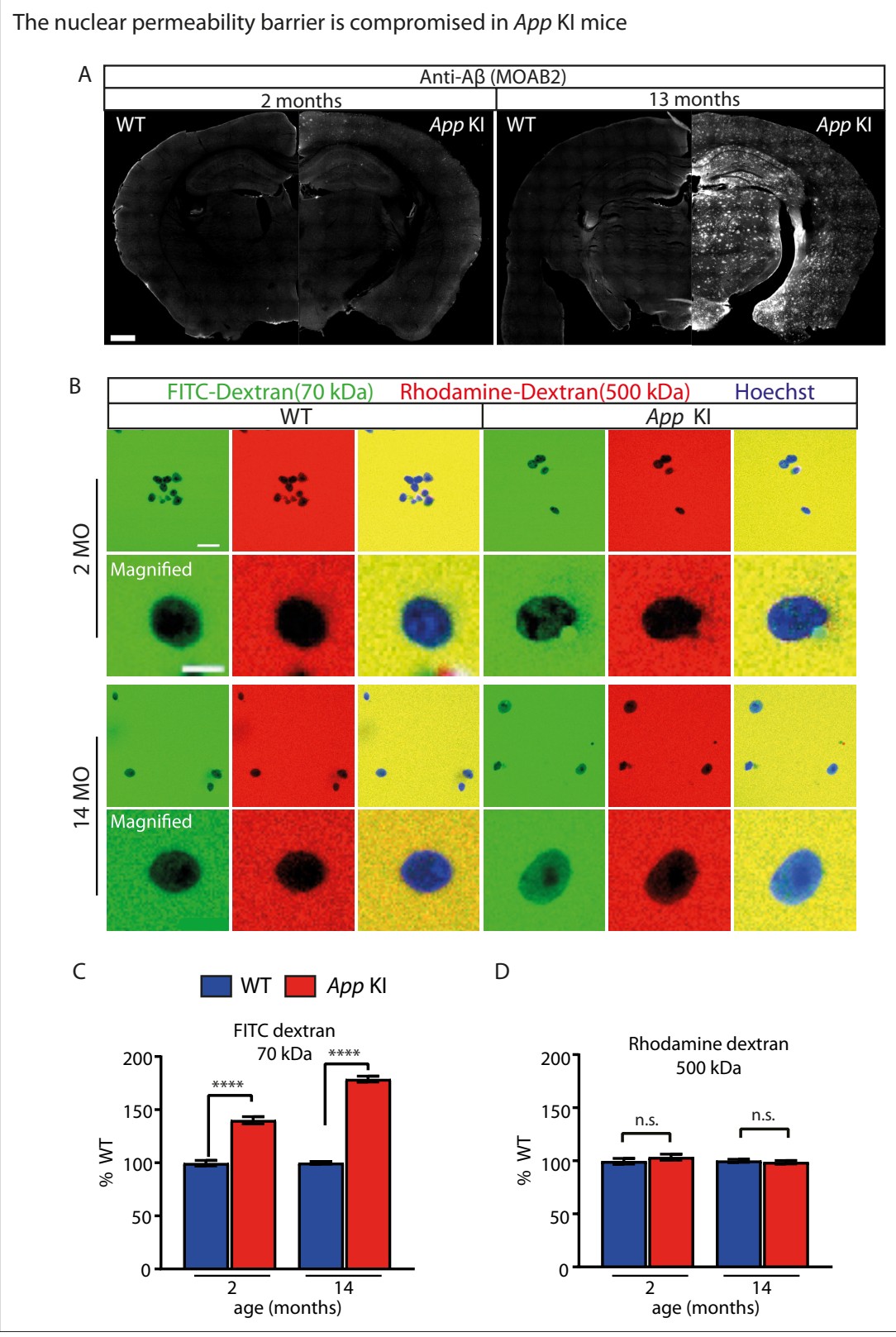

Figure 5. The nuclear permeability barrier is compromised in *App* KI mice. (**A**) Micrograph of 2- and 13 months old WT and *App* KI coronal forebrain sections immunoassayed with MOAB-2 (white). Scale bar is 500 µm. (**B**) Confocal images of purified nuclei from 2- and 14 months WT and *App* KI mice incubated with different sized fluorescence-conjugated dextran. Images show nuclei extracted from 2- and 14 months old animals in the presence of 500 kDa Rhodamine-dextran (red), 70 kDa FITC-dextran (green) and Hoechst nuclear dye (blue). Scale bar at 20 µm. Magnified images show single

*Figure 5 continued on next page*

Figure 5 continued

nuclei. Scale bar at 5 µm. (C-D) Quantification of nuclear intensities for Rhodamine-dextran and FITC-dextran. Group data values shown as percent WTs (n=300 nuclei/group across three replicates). For all graphs values in blue represent WT and values in red represent *App* KI. All data are represented as mean ± SEM. Significance tested by Mann-Whitney U test as follows: not significant (ns), <0.05 (*), <0.01 (**), <0.001 (***), <0.0001 (****). For complete statistical analysis, refer to **Supplementary file 1**.

mechanism (**Jayaraman et al., 2021**; **Caccamo et al., 2017**). Unlike necrosis, necroptosis is a form of programmed cell death triggered when cells are exposed to an inflammatory microenvironment that is enriched with pro-inflammatory cytokines such as tumour necrosis factor alpha (TNF-α). TNF-α is well-linked to the pathophysiology of AD as the levels of TNF-α is significantly elevated in patients and in rodent models of AD (**Jayaraman et al., 2021**; **Caccamo et al., 2017**; **Park et al., 2021**). In cells, binding of TNF-α to its receptors initiates several signaling cascades, one of which leads to the assembly of the necrosome complex, a crucial step in necroptosis. Several reports have shown that key components of the necrosome complex including receptor interacting kinase 3 (RIPK3) and mixed lineage kinase domain-like (MLKL) undergoes nucleocytoplasmic shuttling and is thought to be activated via phosphorylation in the nucleus (**Weber et al., 2018**; **Yoon et al., 2016**). Subsequently, these phosphorylated proteins are exported to the cytoplasm where they contribute toward assembly of the necrosome complex (**Weber et al., 2018**). We postulate that in *App* KI neurons, disruption in the permeability barrier and nucleocytoplasmic transport enhances activation of the component proteins, resulting in an increase in TNFα-induced necroptosis.

To examine if cocultures are susceptible to TNF-α-induced necroptosis, we exposed WT and *App* KI neurons (DIV 14) to a cocktail of drugs (TSZ) used to induce necroptosis (**Weber et al., 2018**.) At 4.5 hr after exposure to TSZ (100 ng TNF-α), the number of WT and *App* KI neurons undergoing necroptosis was elevated, with mutant *App* KI neurons showing higher necroptotic cell death, compared to mock treated controls (**Figure 7A–B**). To ensure that the TSZ cocktail is activating necroptosis, we exposed cocultures to Necrostatin 1 stable (Nec-1s), a specific inhibitor for RIPK1 activity which abolished TNF-α-induced necroptosis (**Figure 7B**). We next performed a dose response curve to see if WT and *App* KI neurons are differentially susceptible to necroptosis when challenged with increasing TNF-α concentrations. As the results indicate, *App* KI neurons recorded a higher fraction of neurons undergoing necroptosis relative to WT controls across all TNF-α concentrations tested (0.1 ng – 100 ng). Moreover, only *App* KI and not WT neurons exhibited significant necroptosis at low TNF-α concentrations (0.1–1 ng), suggesting that *App* KI neurons are more sensitive towards low concentrations of TNF-α (**Figure 7C**). Importantly, under control treatment conditions where no TNF-α is present in the TSZ cocktail, both WT and *App* KI neurons showed comparably low levels of cell death, whereas at the highest level of TNF-α tested in our study (100 ng), *App* KI neurons remained more vulnerable to necroptosis than WT neurons (**Figure 7C**). This suggests that in the absence of external inflammatory stressors such as TNF-α, *App* KI neurons are stable and do not undergo necroptosis.

We next examined the phosphorylation status of MLKL and RIPK3 using phospho-antibodies to label neurons stimulated with TSZ containing either low (0.1 ng) or high (100 ng) concentrations of TNF-α (**Jayaraman et al., 2021**; **Zhang et al., 2021**). Overall, we observed a dose-dependent increase in pMLKL and pRIPK3 in neurons, with high TNF-α stimulated neurons showing stronger levels of phosphorylation as compared to low TNF-α-treated neurons (**Figure 7D–F**). The dose-dependent increase in protein phosphorylation correlates with induction of necroptosis and agrees with the model that phosphorylation of MLKL and RIPK3 are important cellular events prior to necrosome assembly (**Weber et al., 2018**; **Yoon et al., 2016**). We further observed that high concentrations of TNF-α triggered the phosphorylation of the component proteins in both WT and *App* KI neurons when compared to mock-treated (0 ng TNF-α in TSZ) controls while only *App* KI neurons responded to low TNF-α concentrations (**Figure 7E–F**). This provides compelling evidence that *App* KI neurons are more susceptible to TNF-α induced necroptosis through enhanced phosphorylation of MLKL and RIPK3.

To test if neuronal vulnerability toward necroptosis is attributed to loss of NPC function, we asked if inhibiting nuclear transport of proteins would differentially disrupt necroptosis in WT and *App* KI neurons. A previously published experiment in Jurkat cells established that blocking nuclear import and export dampened TNF-α induced necroptosis by preventing the transport and activation of component proteins in the nucleus (**Weber et al., 2018**). We hypothesized that the inhibition of

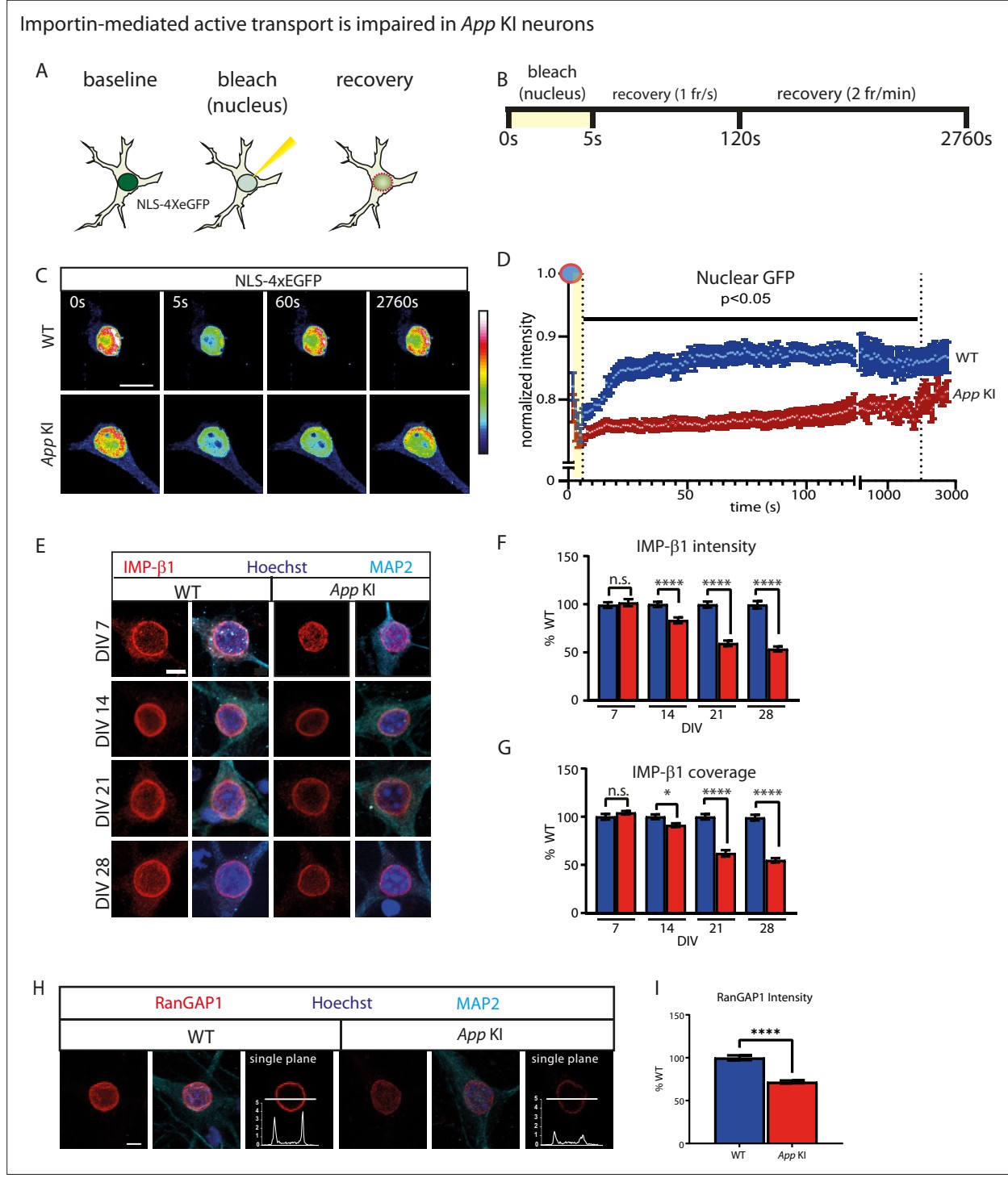

**Figure 6.** Importin-mediated active transport is impaired in *App* KI neurons. (**A**) Schematic of FRAP experiment for neurons expressing NLS-4xEGFP. Partial photobleaching and recovery is performed on the nucleus to track NLS-mediated nuclear import as outlined by the magenta dotted lines. (**B**) Time course for NLS-4xEGFP FRAP experiment and imaging parameters. Image capture was performed at two frame rates: 1 frame/s (t=5–120 s) followed by 2 frames/min (t=121–2760 s). (**C**) Representative images with time stamp (t=0, 5, 60, and 2760 s) of WT and *App* KI neurons during FRAP. Scale bar at 10 μm. (**D**) FRAP recovery curve for NLS-4xEGFP in the nucleus. All data points are normalized against baseline values prior to photobleaching. Shaded region (yellow) signifies photobleaching period. Dotted vertical line indicates all time points between t=6 s and t=1980 s are p<0.05 between WT and App KI (n=10 neurons/group across three replicates: Mann-Whitney U Test). (**E**) WT and *App* KI hippocampal neurons (DIV 7–28) showing Importin β1 (IMP-β1; red) in WT and *App* KI neurons. Cells were co-labeled with MAP2 (cyan), and Hoechst (blue). Scale bar at 5 μm. (**F-G**) Average intensity (**F**) and coverage (**G**) of IMP-β1 in the nuclei. All values are reported as percent WT (n=45 neurons/groups per group across

*Figure 6 continued on next page*

*Figure 6 continued*

three replicates). (**H**) WT and *App* KI hippocampal neurons (DIV 14) showing RanGAP1 (red) nuclear rim localization in WT and *App* KI neurons. A line profile of RanGAP1 signal from a single confocal plane is included for each genotype. Cells were co-labeled with MAP2 (cyan), and Hoechst (blue). Scale bar at 5 μm. (**I**) Average RanGAP1 intensity in the nuclei as in (**H**). All values are reported as percent WT (n=45 neurons/groups per group across three replicates). For all graphs values in blue represent WT and values in red represent *App* KI. All graphs are reported as mean values ± SEM. All significance testing were performed with Mann-Whitney U test: not significant (ns), <0.05 (*), <0.01 (**), <0.001 (***), <0.0001 (****). For complete statistical profiles for each experiment, refer to **Supplementary file 1**.

nuclear transport will be less effective in *App* KI neurons due to the inefficient sequestration of nucleocytoplasmic proteins. To that end, we blocked nuclear transport by incubating cocultures with Leptomycin B (LMB) or with GppNHp. LMB is an inhibitor of CRM-1 mediated nuclear export while GppNHp is a non-hydrolyzable GTP analogue that disrupts the Ran gradient critical for NLS-mediated nuclear import. We tested the efficacy of these drugs in neurons by demonstrating that LMB and GppNHp inhibited nucleocytoplasmic shuttling of NLS-4xEGFP and CRTC1 (***Figure 7—figure supplement 1***; ***Ch'ng et al., 2012***; ***Ch'ng et al., 2015***).

For WT neurons, the inhibition of nuclear export with LMB reduced cell death, while *App* KI saw a similar but smaller reduction (***Figure 7G***). The larger magnitude of decline for WT over *App* KI neurons in necroptosis levels indicates that LMB is less effective in inhibiting cell death in *App* KI neurons (***Figure 7H***). We posit that LMB blockade of nuclear export is less effective due to the compromised permeability barrier allowing activated necrosome proteins to exit the nucleus and accumulate in the cytoplasm. To prove this point, we performed ICC on cocultures and quantified fluorescence intensities of pRIPK3 and pMLKL in different subcellular compartments. The individual fluorescence intensities of both phosphorylated proteins in the soma were unchanged between control and LMB-treated neurons for both WT and *App* KI neurons (***Figure 7I–K***). This is expected since the necroptotic proteins can still enter and become phosphorylated in the nucleus. We next calculated the nuclear to cytoplasmic ratios for pRIPK3 and pMLKL separately and found both proteins enriched in the nucleus of only LMB-treated WT but not *App* KI nuclei (***Figure 7L***), thus validating our hypothesis that the compromised permeability barrier is preventing efficient retention of phosphorylated proteins in the nucleus and allowing the proteins to enter the cytoplasm.

While LMB hinders necroptosis by blocking activated proteins from exiting the nucleus, GppNHp reduces necroptosis by inhibiting nuclear entry and subsequent phosphorylation of RIPK3 and MLKL (***Weber et al., 2018***), a process that is also likely to be impaired if the nuclear barrier is compromised. Indeed, incubation with GppNHp resulted in a reduction in necroptosis only in WT neurons (***Figure 6G–H***). ICC of TSZ-stimulated neurons treated with GppNHp showed that in WT neurons, GppNHp significantly reduced the somatic levels of pRIPK3 and pMLKL (***Figure 7J–K***) while no reduction was observed for pRIPK3 in *App* KI neurons (***Figure 7J***). However, there is a partial reduction of pMLKL in *App* KI neurons (***Figure 7K***). Overall, the differences observed between WT and *App* KI neurons supports our hypothesis that the efficacy of GppNHp in reducing phosphorylation of RIPK3 and MLKL is lower in *App* KI neurons due to the compromised permeability barrier which allows unregulated entry of non-phosphorylated proteins into the nucleus (***Figure 7—figure supplement 2***).

In summary, these experiments collectively link the loss of NUPs and degradation of the nuclear permeability barrier in *App* KI neurons with an increase in neuronal vulnerability toward TNF-α induced necroptosis.

## Discussion

### Loss of nucleoporins disrupts NPC function and impacts cellular processes

In this report, we demonstrate an Aβ-driven loss of NUP expression in *App* KI hippocampal neurons both in primary cocultures and in the mouse model. High-resolution confocal microscopy further revealed that the reduction of NUP expression corresponded with fewer NPC puncta that are distributed further apart on the nuclear envelope, thus indicating a potential loss of NPCs in *App* KI neurons. While pan-NPC antibodies frequently detect multiple NUP subunits, they have been used to identify individual complexes especially with high-resolution confocal or electron microscopes that are able to resolve these structures. Nevertheless, future experiments to visualize the nuclear membrane will be

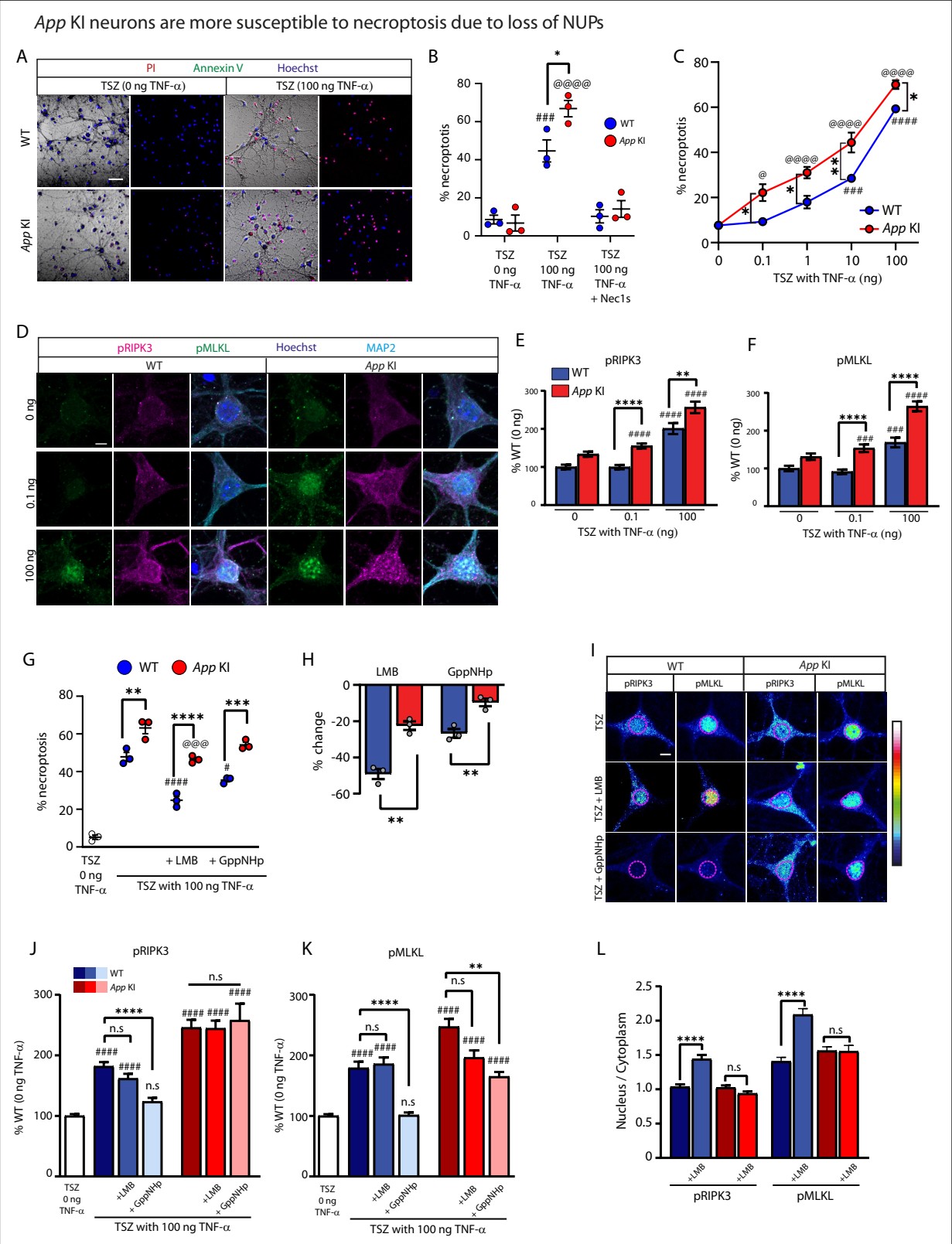

*App* KI neurons are more susceptible to necroptosis due to loss of NUPs

**Figure 7.** *App* KI neurons are more susceptible to necroptosis due to loss of NUPs. (**A**) Representative live imaging bright field image of WT and *App* KI neurons exposed to TSZ cocktail containing 0 or 100 ng TNF- α. Cells stained with Hoechst (blue), propidium iodide (red), and FITC-Annexin V (green). Cells with red are necroptotic cells, cells with green signal are apoptotic cells, brightfield used to identify neurons for quantification. Scale bar at 50 µm. (**B**) Quantification of necroptotic cell of WT and *App* KI neurons exposed to TSZ cocktail with 0 or 100 ng TNF-α in the presence of Necrostatin

*Figure 7 continued on next page*

*Figure 7 continued*

1 stable (Nec1s; N=3 replicates; [#] WT TSZ 0 ng v. WT TSZ 100 ng; [@] *App* KI TSZ 0 ng v. *App* KI TSZ 100 ng; [*] WT TSZ 100 ng v. *App* KI TSZ 100 ng). (**C**) Concentration curve of WT and *App* KI neurons exposed to increasing concentrations of TNF-α (0–100 ng) showing percent necroptotic cell death (N=3 replicates; [#] significance tests for WT TSZ 0 ng v. WT TSZ TNF-α; [@] significance tests for *App* KI TSZ 0 ng v. *App* KI TSZ TNF-α; [*] significance tests for WT TSZ TNF-α v. *App* KI TSZ TNF-α for individual concentrations). (**D**) ICC of WT and *App* KI neurons exposed to TSZ cocktail with 0–100 ng TNF-α stained for pRIPK3 (magenta), pMLKL (green), MAP2 (cyan), and Hoechst (dark blue). Scale bar at 5 μm. (**E-F**) Quantification of somatic pRIPK3 (**E**) and pMLKL (**F**). Values reported as percent change from WT- 0 ng TNF-α (n=45 neurons/group across three replicates; [#] significance tests for all groups v. WT TSZ 0 ng TNF-α; [*] significance tests for WT TSZ TNF-α v. *App* KI TSZ TNF-α samples at specific TNF-α concentrations). (**G**) Quantification of percent necroptotic cell death for WT and *App* KI neurons exposed too TSZ in the presence of LMB (50 nM; 2 hr) or GppNHp (500 nM; 2 hr) (N=3 replicates; [#] significance tests for WT groups v. WT TSZ 100 ng TNF-α; [@] significance tests for all *App* KI groups v. *App* KI TSZ 100 ng TNF-α; [*] significance tests for WT TSZ TNFα+LMB/GppNHp v. *App* KI TSZ TNF-α+LMB/GppNHp). (**H**) Graph showing percent change in cell death for WT and *App* KI neurons with LMB and GppNHp compared to their respective 100 ng TNF-α controls (N=3 replicates; [*] WT TSZ TNFα+LMB/GppNHp v. App KI TSZ TNF-α+LMB/GppNHp). Significance tested by unpaired t-test. (**I**) Representative pseudo-colored confocal images and heatmap of neurons exposed to TSZ in the presence of LMB and GppNHp and labeled with pRIPK3 and pMLKL. Nuclei outlined in dotted magenta shapes Scale bar at 5 μm. (**J-K**) Somatic quantifications of pRIPK3 (**J**) and pMLKL (**K**) from I (n=45 neurons/group across three replicates; [#] significance tests for all groups v. TSZ TNF-α 0 ng; [*] significance tests for indicated comparisons between groups). (**L**) Nucleus to cytoplasmic ratio for neurons exposed to LMB showing nuclear accumulation of pRIPK3 and pMLKL in WT neurons. Significance testing performed by Mann Whitney U test (n=45 neurons/group across three replicates). For all graphs, values in blue represent WT and values in red represent *App* KI. All graphs are reported as mean values ± SEM. Significance testing with two-way ANOVA (Tukey; B, C, G) or one-way ANOVA (Kruskal-Wallis; E, F, J, K) unless otherwise stated. For all statistics significance is as follows: not significant (ns), <0.05 (*), <0.01 (**), <0.001 (***), <0.0001 (****). For complete statistical profiles for each experiment, refer to ***Supplementary file 1***.

The online version of this article includes the following figure supplement(s) for figure 7:

**Figure supplement 1.** LMB blocks nuclear export and GppNHp inhibits nuclear import in hippocampal neurons.

**Figure supplement 2.** App KI neurons are more susceptible to inflammation-induced necroptosis.

crucial for characterizing the extent and nature of the damage in NPCs. In short, our model suggests that a reduction of NUPs in *App* KI neurons leads to NPC dysfunction and a degraded permeability barrier which results in an impairment of nucleocytoplasmic protein compartmentalization and a disruption of active transport of NLS-tagged proteins (***Figure 8***).

NPCs are large assemblies consisting of several hundred individual proteins (***Alber et al., 2007***) and in this study, we chose to examine NUP107 as it is a long-lived NUP that forms part of a major scaffold subassembly of the NPC known as Y-NUPs (***Savas et al., 2012***; ***D'Angelo et al., 2009***). While the loss of a single NUP may not result in a total collapse of the nuclear pore architecture and function, several studies have shown that depletion of individual components of the Y-NUPs (i.e. NUP107, NUP160, NUP133, etc.) can trigger a domino effect on other NUPs, thereby reducing their concentrations, overall NPC density, and disrupting protein and RNA transport (***Harel et al., 2003***; ***Walther et al., 2003***; ***Boehmer et al., 2003***; ***Souquet et al., 2018***; ***Siniossoglou et al., 1996***). Unlike Y-NUPs, NUP98 is a peripheral FG-repeat subunit that resides in the central channel of the NPCs as well as in the cytoplasm (***Griffis et al., 2003***). Even though NUP98 is not a long-lived protein, it remains an essential component of the NPC as a knockout mouse displayed deficits in active transport of nuclear proteins and inefficient assembly of selected NUPs in the NPC (***Wu et al., 2001***). Our interest in studying NUP98 is also partially motivated by reports that Tau directly interacts with NUP98, causing mislocalization of the protein in NFTs and disrupting nuclear transport (***Eftekharzadeh et al., 2018***). Many of these studies on NUP107 and NUP98 are in agreement with our experiments in *App* KI neurons showing that loss of these selected NUPs, as detected by both pan-NPC and individually-validated antibodies, correspond to lower NPC counts and deficits in nucleocytoplasmic compartmentalization and transport.

It should be noted that NUPs also participate in other cellular processes which may be perturbed in *App* KI neurons (***Strambio-De-Castillia et al., 2010***). For example, NUP98 and NUP107 can regulate transcription by a variety of means (***Cho and Hetzer, 2020***). NUP107 forms part of the outer ring sub-complex that tethers the genome and preferentially associates with active promoters (***Gozalo et al., 2020***), while NUP98 can function as a transcription modulator in the nucleoplasm by interacting with a variety of binding partners for gene activation and silencing, as well as being a tethering point for importins (***Cho and Hetzer, 2020***). In addition, NUPs have also been shown to perform extra-nuclear function such as assembly and function of primary cilia (***Obado and Rout, 2016***; ***Endicott***

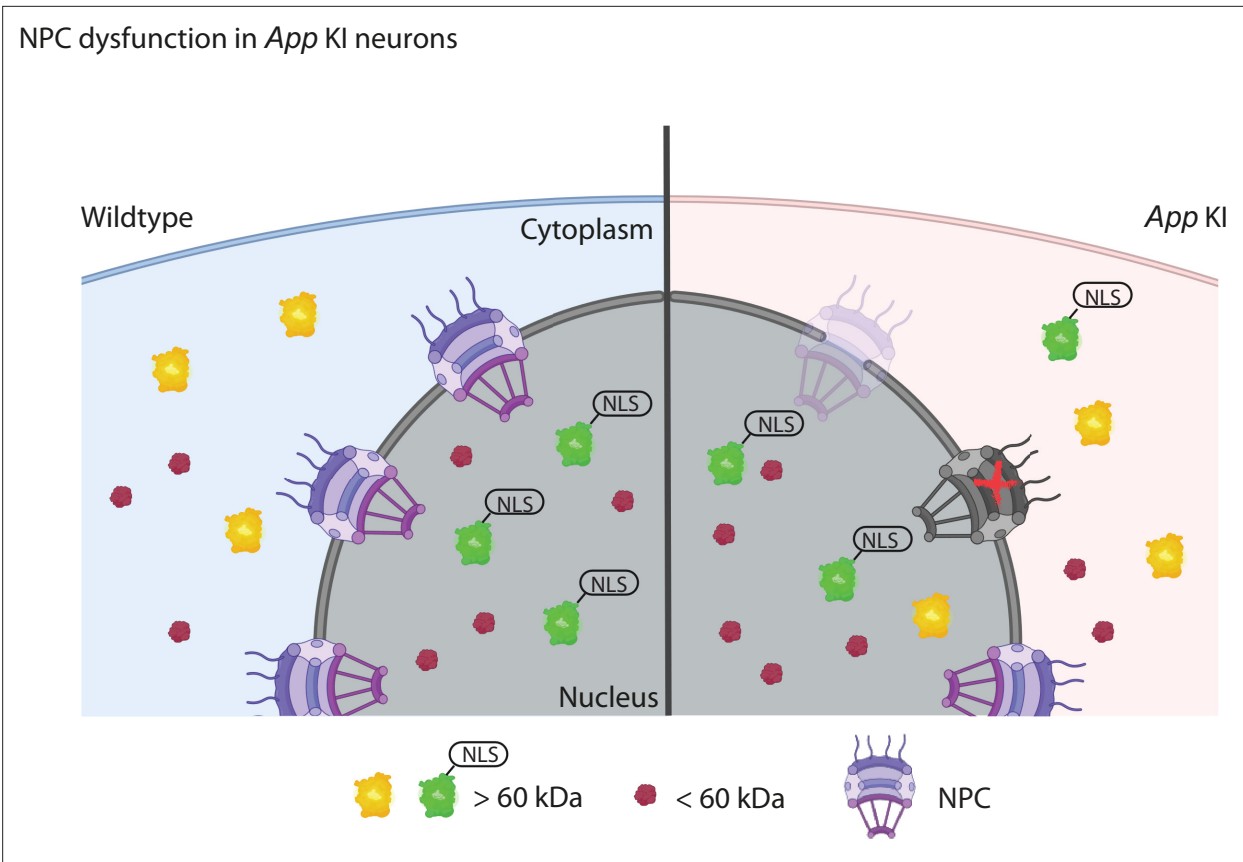

**Figure 8.** NPC dysfunction in *App* KI neurons. A diagram modeling the reduction of NUPs and potential loss of NPCs in *App* KI neurons contributing toward a breakdown of the permeability barrier and disruption of nucleocytoplasmic compartmentalization and protein transport. Localization of proteins <60 kDa (magenta) or >60 kDa, with (green) or without (yellow) a nuclear localization sequence (NLS) are altered in *App* KI nuclei.

*and Brueckner, 2018*). It will be of interest to examine if NPC dysfunction in *App* KI neurons extends beyond regulating the permeability barrier and nuclear transport.

## Accumulation of intracellular Aβ correlates with nuclear pore dysfunction in *App* KI neurons

We show evidence the expression and subsequent cleavage of mutant APP to generate Aβ is required for the reduction of NUPs in neuronal nuclei. Inhibition of γ-secretase activity prevented cleavage of mutant APP and generation of Aβ, which led to the partial restoration of NUP levels. Direct incubation of oligomeric, but not monomeric or fibrillar Aβ42 in WT neurons also resulted in the loss of NUPs. We also consistently observed that levels of intracellular Aβ correlates with the amount of NUP reduction in *App* KI neurons. However, it is unclear if different fragments of cleaved APP or different forms of Aβ, apart from Aβ42, also contribute toward the NPC dysfunction as both APP-CTF and Aβ40 can be detected in *App* KI mice (*Saito et al., 2014*). Taken together, these experiments indicate that Aβ is the primary driver for the loss of NUPs and there is a strong likelihood that accumulation of intracellular Aβ is partially responsible for causing this phenotype. What is less understood is how, and what form of Aβ accumulation inside the cell interacts with NUPs. At 2 months in *App* KI mice, extracellular plaques are sparsely distributed, yet we readily detect a more diffuse staining of intracellular Aβ (*Figure 3B*), a loss of NUPs, and a degradation of the permeability barrier. In older *App* KI mice, Aβ-positive amyloid plaques are found in abundance throughout the hippocampus and the intracellular Aβ signals are more punctate (*Figure 5A*; *Figure 3—figure supplement 1*). In agreement with our observations, studies in humans, primates, and rodent models of the disease have demonstrated intracellular Aβ accumulation in AD brains (*Iulita et al., 2014*; *LaFerla et al., 1997*; *Oddo et al., 2003b*; *Kimura et al., 2003*; *D'Andrea et al., 2001*; *Wirths et al., 2001*; *Takahashi et al., 2004*), arguing for a key role of intracellular

Aβ in causing the nuclear pore dysfunction during early stages of the disease. Some of these molecular studies have further identified Aβ accumulation in late endosomes and multivesicular bodies that precede plaque deposition, arguing that this phenomenon is an early event in pathogenesis (*Gouras et al., 2000*; *Takahashi et al., 2002*; *Oddo et al., 2003a*; *Knobloch et al., 2007*). What is less clear from these studies is the nature of intracellular Aβ that trigger the cellular damage. Recent studies from *Lee et al., 2022* using several mouse models of AD showed faulty Aβ-laden autophagic vacuoles that coalesce around the nucleus, causing the formation of membrane tubules and intraluminal accumulation of fibrillar β-amyloid in the perinuclear space (*Lee et al., 2022*). An earlier study also detected the presence of Aβ and APP-derived fragments in the nucleus and perinuclear compartment (*Pensalfini et al., 2014*). While the assembly of these autophagic vacuoles was not examined in *App* KI animals, it is possible that the NPC deficits in *App* KI is linked to accumulation of β-amyloid around the perinuclear space. While we often observed Aβ-positive punctate structures adjacent to nuclei, it is remains to be determined whether there is direct association of Aβ – in whichever form – with NPCs.

Another aspect of intracellular Aβ that is unresolved is the origin or source of Aβ that accumulates inside the cell that leads to NPC dysfunction. One possibility is the extracellular uptake of Aβ into the cell that could damage the nuclear pores (*Lai and McLaurin, 2010*). We briefly tested this possibility by performing a media swap experiment and saw that Aβ-laden conditioned media derived from *App* KI neurons increased intracellular Aβ and decreased NUP levels in WT neurons. While extracellular Aβ is internalized as endolysosomes and initially precluded from the nucleus, several studies have shown that Aβ-induced lysosomal leakage from these vesicular structures could release the contents of the lysosomes which includes Aβ and other proteases that may cleave NUPs (*Ditaranto et al., 2001*; *Ji et al., 2002*; *Yang et al., 1995*; *Yang et al., 1998*). Apart from endocytosis, other unconventional routes of entry have also been proposed, including a direct, energy-independent uptake of soluble Aβ via passive diffusion (*Kandimalla et al., 2009*). Finally, it is important to note that our experiments do not rule out that extracellular Aβ, whether internalized or binding to cell surface receptors, can also indirectly impact NPCs by triggering damaging cellular cascades (*Takuma et al., 2009*; *Benilova et al., 2012*). Future experiments designed to study how Aβ and NPCs interact with neurons will shed light on the mechanism of NPC dysfunction.

## Age-dependent decline in nuclear function during normal aging and in AD

Increasingly, nuclear pore dysfunction has been reported in many aging-related neurodegenerative disorders (*Li and Lagier-Tourenne, 2018*). While the molecular mechanism of the dysfunction is distinct for each disorder, the general disruption in NPC architecture, permeability barrier, nucleocytoplasmic transport of macromolecules, and transcriptional outputs share overlapping characteristics. While there is increasing evidence that proteinopathies interfere with NPC function, the longevity and stability of NPCs and nuclear transport also declines during normal aging process, with several cell and animal model studies showing an age/time-dependent reduction in NPCs and nuclear adaptor protein complexes, or loss of nuclear integrity (*Mertens et al., 2015*; *D'Angelo et al., 2009*; *Rempel et al., 2019*; *Toyama et al., 2019*). Our experimental data hints that the decline in nuclear pore integrity also occurs in WT neurons, and is accelerated in *App* KI neurons. Intriguingly, reports have shown an increase of intracellular Aβ in aged primary WT neurons, suggesting the possibility that the normal decline of NPC function may be linked to gradual Aβ accumulation during the normal aging process (*Burrinha et al., 2021*; *Guix et al., 2012*; *Baker-Nigh et al., 2015*). As age is the biggest risk factor for many types of dementia, we speculate that a general decline in nuclear function, because of normal aging, and exacerbated in the presence of Aβ, contributes toward neuronal loss and the emergence of clinical pathologies associated with AD. However, the temporal sequence of this phenomenon is not yet established. We cannot distinguish whether nuclear pore dysfunction is a cause or symptom of the disease. In all likelihood, as cellular mechanisms undergo age-dependent decline and cellular homeostasis is compromised, the loss of nuclear integrity could promote downstream AD-associated pathologies. However, it is equally possible that the convergence of risk factors including the continuous, low-level presence of intracellular Aβ during the cellular phase of the disease accelerate nuclear pore dysfunction which may, in turn, promote the emergence of AD pathologies (*De Strooper and Karran, 2016*). The second scenario is particularly relevant to long-lasting damage sustained to NPCs as studies have shown that several NPC subunits are extremely long-lived proteins with low turnover

rates in mammalian brains and in cultured cells (*Savas et al., 2012*; *Toyama et al., 2013*; *Cho and Hetzer, 2020*; *Mathieson et al., 2018*; *Dörrbaum et al., 2018*; *Rabut et al., 2004*). In addition to its longevity, mRNA expression of select NUPs, specifically a subset of the Y-NUPs that form part of the NPC scaffold are downregulated in adult worms and rodents. Incredibly, NUP107 and NUP153 RNA is dispensable in adult worms as eliminating its expression did not impact viability (*D'Angelo et al., 2009*). These studies collectively indicate that there is minimal NPC biogenesis in non-mitotic cells as replacing the core scaffold proteins in nuclear pores would require a complete disassembly of the entire NPC complex, something that regularly occurs in mitotic cells when the nuclear membrane is remodeled during cell division (*De Magistris and Antonin, 2018*; *Morchoisne-Bolhy et al., 2015*). The fact that NPCs are stable structures with low turnover rates highlights a unique vulnerability for AD neurons: If Aβ expressing neurons accrue minor insults to the NPCs over time, the damage cannot be rapidly restored, and the cumulative impairments could eventually jeopardize cellular homeostasis with devastating consequence such as neuronal loss. Indeed, some of our experiments using the mouse model hint at the possibility that subtle nuclear deficits already occur at very initial stages of AD.

Nevertheless, a few outstanding questions remain from our study. Even though we observed loss of selected NUPs and a degraded permeability barrier in young animals, it is unclear to what extent neurons can withstand damage to the NPCs before it becomes irreversible. As neuronal loss is a devasting outcome in mammalian brains, cellular homeostasis along with feedback, and feed forward mechanisms must provide a constant check and balance to prevent further deterioration of the nucleus and induction of an irrevocable event such as necroptosis (*Figure 7—figure supplement 2*). Along with intrinsic cellular factors, changes in the extracellular milieu such as degree of neuroinflammation, accumulation of Aβ plaques or even altered glial responses – some of which are absent in coculture systems – are additional factors that may influence severity of the nuclear pore dysfunction and necroptosis. Apart from altered cellular homeostasis, it is important to understand how Aβ and Tau interact to damage the NPC. While both proteinopathies contribute toward nuclear pore deficits, it is unclear if the presence of both proteinopathies amplify the damage to NPCs. It is possible that progressive insults by Aβ that accelerates NPC dysfunction disrupts cellular homeostasis and promotes emergence of tau tangles. Future studies should examine how the temporal sequence in which Aβ and Tau pathologies manifests in AD brains shape the type and degree of damage to the nucleus during progression of the disease.

To summarize, this study along with others, continue to demonstrate common features of nuclear pore dysfunction, disruption in permeability barrier and nuclear transport in age-related neurodegenerative diseases. While finding a single drug target for a multi-subunit protein complex such as NPCs is challenging, shoring up common mechanisms regulated by NPCs such as repairing the nuclear permeability barrier, preventing degradation of NPCs, or enhancing adaptor protein-based nuclear import and export might be a more amenable therapeutic approach.

# Materials and methods
## Animals
All animal work was conducted in the Animal Research Facility (ARF) at Lee Kong Chian School of Medicine, Nanyang Technological University, Singapore, following protocols approved by Nanyang Technological University Institutional Animal Care and Use Committee (IACUC protocol A18091 and A18095). The *App*^NL-G-F/NL-G-F^ (*App* KI) mouse was obtained from T. Saido (*Sasaguri et al., 2017*; *Saito and Saido, 2018*) while C57BL/6 (WT) strains were bred from colonies obtained from the animal facility or purchased from InVivos Pte. Ltd (Singapore).

## Antibodies and plasmids

| Primary Antibodies | Source | Dilutions |
| --- | --- | --- |
| Amyloid β (82E1; mouse) | IBL America (10323) | 1:1000 (ICC) |
| Amyloid β (MOAB-2, 6C3) | Abcam (MABN254) | 1:1000 (ICC, IHC) |

*Continued on next page*

*Continued*

| Primary Antibodies | Source | Dilutions |
|---|---|---|
| CRTC1 (rabbit) | Bethyl Laboratories (A300769A) | 1:1000 (ICC) |
| GFP (rabbit) | Thermo Fisher Scientific (A-11122) | 1:1000 (ICC) |
| H3(rabbit) | Abcam (ab1791) | 1:1000 (IB) |
| Importin-β1 (3E9; mouse) | Thermo Fisher Scientific (MA3-070) | 1:1000 (ICC) |
| LAMIN-B1 (rabbit) | Abcam (ab16048) | 1:1000 (ICC, IB) |
| MAP2 (chicken) | PhosphoSolutions (1100-MAP2) | 1:500 (ICC), 1:200 (IHC) |
| MLKL phospho S345 (ERP9515(2); rabbit) | Abcam (ab196436) | 1:1000 (ICC) |
| NPC (RL1; mouse) | Santa Cruz Biotechnology (sc-58815) | 1:1000 (ICC), 1:500 (IHC) |
| NUP107 (39C7; mouse) | Thermo Fisher Scientific (MA1-10031) | 1:1000 (ICC), 1:250 (IHC), 1:200 (IB) |
| NUP98 (2H10; rat) | Abcam (ab50610) | 1:1000 (ICC, IB), 1:500 (IHC) |
| RanGAP1 (19C7; mouse) | Thermo Fisher Scientific (33–0800) | 1:1000 (ICC) |
| RIP3 phospho T231 +S232 (2D7; mouse) | Abcam (ab205421) | 1:1000 (ICC) |
| Tau phosphor Ser202, Thr205 (AT8; mouse) | Thermo Fisher Scientific (MN1020) | 1:1000 (ICC) |
| **Secondary Antibodies** | **Source** | **Dilutions** |
| Alexa Fluor 647 goat anti-chicken IgY (H+L) | Invitrogen (A-21449) | 1:1000 (ICC), 1:500 (IHC) |
| Alexa Fluor 488 goat anti-rabbit IgG (H+L) | Invitrogen (A-11034) | 1:1000 (ICC, IHC) |
| Alexa Fluor 555 goat anti-mouse IgG (H+L) | Invitrogen (A-21422) | 1:1000 (ICC, IHC), 1:500 (only for IHC NUP107) |
| FITC donkey anti-rat IgG (H+L) | Invitrogen (A-18740) | 1:1000 (ICC), 1:500 (IHC) |
| VeriBlot for IP detection reagent (HRP) | Abcam (ab131366) | 1:1000 (IB) |
| **Dyes** | **Source** | **Dilutions** |
| Hoechst 33342 | Thermo Fisher Scientific (H3570) | 1:1000 (IHC, cell death assay) |
| Propidium Iodide | Thermo Fisher Scientific (P1304MP) | 1:1000 (cell death assay) |

All GFP constructs used in FRAP experiments have been reported (*Ch'ng et al., 2015*). Unless otherwise stated, all animal experiments were conducted in age-matched female mice ranging from 2- to 18 months old. Hippocampal neuron cocultures were prepared from post-natal (P0-P2) WT and *App* KI mouse pups.

## Primary mouse hippocampal cocultures and transient transfections

The mouse hippocampus was isolated following a previously described protocol (*Ch'ng et al., 2012*). In brief, hippocampi from WT and *App* KI pups (P0-P2) were isolated in ice-cold HBSS (1 mM sodium pyruvate, 1% HEPES; Gibco) and trypsinized (0.25% trypsin, 1.26 mM CaCl2, and 0.48 mg/ml DNase in HBSS) for 15 min followed by incubation with an inhibitor solution for 5 min (1 mg/ml trypsin inhibitor in HBSS). The tissue was gently triturated 10 x in plating media (2% B27, 1% GlutaMAX, 0.01% FBS, and 200 ng/mL Gentamycin in Neurobasal medium). On average, $6.5 \times 10^4$ cells were seeded on to individual Poly-DL-Lysine (25–40 kDa P-DL-L; 0.1 mg/ml; Sigma) coated 12 mm glass coverslips or P-DL-L coated 8-well NunC Chambers (Thermo Fisher Scientific). Cells were incubated with plating media for 24 hr before being swapped with feeding media (2% B27 and 1% GlutaMAX in Neurobasal medium) and maintained for a maximum of 28 days in vitro (DIV). All steps pre-plating were conducted with solutions heated to 37 °C. Cocultures were incubated in a humidified incubator at 37 °C and 5% $CO_2$ at all times. Unless otherwise stated, all calcium phosphate transfections, were performed on DIV13 cocultures using a high calcium-phosphate protocol (*Jiang and Chen, 2006*). In brief, prewarmed Neurobasal A media was added to replace conditioned media in cocultures at least 1 hr prior to transfections. DNA-calcium precipitates were formed by combining Solution A (1 μg cDNA, 0.31 M $CaCl_2$) with Solution B (2 x HeBS - 0.27 M NaCl, 9.52 mM KCl, 1.42 mM $Na_2HPO_4 \cdot 7H_2O$, 14.99 mM D-glucose, 0.04 M HEPES, adjusted to 7.05 pH with NaOH) at a 1:1 ratio and incubated in the dark. The precipitates were then added to cocultures and incubated at 37 °C and at 5% $CO_2$

for a variable amount of time depending on the plasmid constructs expressed. Once incubation is completed, coverslips were rinsed with Neurobasal medium and replaced with conditioned media. Cocultures were used at DIV14 for live cell imaging.

## Perfusion, cryo-sectioning, and preservation of mouse brains

Age-matched WT and *App* KI mice were anesthetized by intraperitoneal injection of pentobarbital in accordance with standard animal procedures approved by ARF. Mouse reaction was intermittently assessed by paw pinch until they displayed no reaction. Mice were perfused with ice-cold PBS followed by 4% PFA in 0.1 M phosphate buffer. Brains were immediately harvested and post fixed in 4% PFA at 4 °C and dehydrated in increasing concentrations of sucrose (10–30%) in PBS over the next 48 hr at 4 °C. Once adequately dehydrated, brains were frozen on a cryostat chuck and covered in embedding compound (OCT; VWR chemicals). Cryostat (Leica) was maintained at –20 °C, and coronal brain sections were taken at 40 µm thickness, and stored in cryoprotective solution (30% ethylene glycol, 30% glycerol, in 20 mM PB) at –20 °C. APP/PS1 with age-matched WT sections were acquired from Sreedharan lab (*Navakkode et al., 2021*). Briefly, hippocampal brain sections were prepared using a vibratome (4 °C at 100 µm). Sections were immediately fixed with 4% PFA overnight at 4 °C and stored at –20 °C in cryoprotectant until processed for immunohistochemistry (IHC).

## Immunoassays

For immunocytochemistry (ICC), hippocampal cocultures were fixed with 4% PFA/PBS, permeabilized in 0.1% Triton-X/PBS and blocked in 10% goat serum/PBS before primary antibody incubations. Unless otherwise stated, all antibody dilutions were done in blocking solution and primary antibody incubations performed for 3 hr, and secondary antibody +Hoechst incubation for 1 hr at room temperature (RT). Between each step coverslips were washed 3 X with gentle agitation in PBS. All coverslips were mounted using Aqua-Poly/Mount (PolySciences). For IHC, perfused brain sections stored in cryoprotective media were washed three times in 0.1%Triton-X/PBS and permeabilized in 0.2% Triton-X/PBS. Sections were placed in blocking solution (0.1% Triton-X, 10% Goat serum in PBS) for 1 hr and incubated with primary antibody overnight at 4 °C and secondary antibody +Hoechst for 2 hr at RT. All antibody incubations were diluted in blocking solution and brain sections were washed in 0.1% Triton-X/PBS between incubations. Stained sections were mounted onto Poly-lysine adhesion slides (Polysine) with Aqua-Poly/Mount. All steps were conducted at RT unless otherwise stated with gentle agitation. For immunoblotting (IB) of homogenates and nuclear lysates, 10 µg of samples and ladder were loaded onto a 12% resolving gel (Bio-Rad) and SDS-PAGE and western blot transfers were performed using standard protocols. Unless otherwise stated, all blots were incubated in primary antibodies at 4 °C overnight and secondary antibodies for 1 hr at RT. All blots were washed in TBST buffer (20 mM Tris, 150 mM NaCl, 0.2% Tween 20) in RT. Blots were incubated with either SuperSignal West Pico PLUS Chemiluminescent Substrate or SuperSignal West Femto Maximum Sensitivity Substrate (Thermo Fisher Scientific), imaged using ChemiDocMP imaging system and analysed using Image Lab software (Bio-Rad). All quantification was performed using Image Lab with band intensities determined within the same area using the volume tool and normalized against H3 as a loading control.

## Cell viability assays

Apoptosis and necrosis were assessed for DIV 14 cocultures using a kit (Biotium) following manufacturer protocols. In brief, coverslips were incubated in staining solution (FITC-Annexin V, Ethidium Homodimer III, and Hoechst in 1 X binding buffer) for 15 min at RT. As positive controls, WT coverslip were incubated in 500 µM $H_2O_2$ (Sigma-Aldrich) for 12 hr prior to staining. Cells were imaged live in Tyrode's buffer supplemented with Trolox (140 mM NaCl; 10 mM HEPES pH 7.3; 5 mM KCl; 3 mM $CaCl_2$; 0.1 mM $MgCl_2$; 10 mM glucose; 10 nM Trolox pH 7.35). Images were taken from five quadrants from across the coverslip (top, bottom, left, right, centre) using a 20 X air objective. For apoptosis and necrosis assay nuclear intensities were calculated from a surface based on Hoechst. Thresholds for positive signal were based on the H2O2-positive control.

## Nuclear extraction and fluorescent-dextran assay

Age-matched WT and *App* KI mice were anesthetized using isoflurane and euthanized via cervical dislocation. Whole forebrain was harvested and homogenized in a 5 ml Potter-Elvehjem tissue grinder

with homogenization buffer (HB; 0.25 M sucrose, 25 mM KCl, 5 mM MgCl2, 20 mM tricine, pH 7.8) containing protease inhibitor cocktail (Roche) for 10 strokes. 0.3% IGEPAL-CA 630 was added to the homogenate and ground for five additional strokes. Homogenates were filtered through a 40 µm strainer and mixed with sucrose cushion buffer (SCB; 1.8 M sucrose, 10 mM tris-HCl, 1.5 mM MgCl$_2$, pH 6.9) at a ratio of 1:2.3 before layering on 3 ml of SCB in a 12 ml Thinwall Ultra-clear centrifuge tubes (Beckman Coulter). Samples were centrifuged at 30,000 x $g$ for 45 min at 4 °C using a SW41 Ti swinging bucket rotor (Beckman Coulter) before the supernatant was decanted, and the nuclear pellets resuspended in SCB. A mixture containing 70 kDa FITC-Dextran (0.1 g/ml; Sigma-Aldrich), 500 kDa rhodamine dextran (0.1 g/ml; Sigma-Aldrich) and Hoechst was mixed in a 10:10:1 ratio in PBS. This mixture was then added to the nuclear suspension for 30 min on ice at a 10:1 ratio. A droplet of the nuclear suspension was spotted onto a Polysine adhesion glass slide for imaging. For all coverslips, a total of five images were obtained, one in each designated quadrant (4 corners and the middle). For western blots of nuclear isolates, the suspension was lysed in RIPA buffer, and 10 µg of proteins was loaded per sample for SDS-PAGE.

## Aβ coculture experiments

All Aβ coculture experiments were replicated a minimum of three times. Pharmacological treatments were performed on cocultures at various stages of maturity. Unless stated, all reagents and treatments were made with conditioned media from corresponding cocultures. For DAPT experiments, 50% media change was performed 3 consecutive days from DIV7-10 to reduce extracellular Aβ. From DIV10-14, DAPT (10 µM in DMSO) was added to the media at 24 hr intervals along with DMSO-only mock-treated controls. At DIV14, cocultures were processed for ICC. For synthetic Aβ42 experiments, Aβ42 monomers, oligomers, and fibrils were prepared fresh for each experiment from monomeric Aβ42 stock following a well-established and widely used protocol (*Pan et al., 2011*). In brief, for Aβ42 oligomers preparations, monomers (100 µM) were diluted in DPBS, sonicated in a water bath for 5 min at RT and incubated at 37 °C for 2 hr followed by another 24 hr at 4 °C. The preparation was then spun at 5000 x $g$ for 50 min at 4 °C to form a pellet which was resuspended in media at 500 nM and added to cocultures at a final concentration of 50 nM. Aβ42 fibrils were prepared in a similar manner, except the 37 °C incubation step was extended to 1 wek. The Aβ42 preparations were added to cocultures at DIV10 and processed for ICC at DIV14. For media exchange experiments between WT and *App* KI cocultures, DIV19 conditioned media were harvested from each well, pooled and redistributed among the groups. Cocultures exposed to the exchanged media we processed for ICC at DIV21.

## TNF-α necroptosis assay

For TNF-α induced necroptosis, cocultures were stimulated with TSZ cocktail [TNF-α (0–100 ng), SMAC (2 µM), z-VAD (10 µM)] with variable concentrations of TNF-α depending on experimental needs (*Zhu et al., 2018*). To inhibit necroptosis, cocultures were treated with Necrostatin-1 stable (Nec-1s, 2 µM) for 30 min prior to TSZ stimulation (*Caccamo et al., 2017*). To inhibit nuclear export and import, cocultures were pre-treated with Leptomycin B (LMB, 50 nM, Invivogen) or GppNHp (non-hydrolyzable GTP analogue, 500 nM, Abcam) respectively for 2 hr prior to and during TSZ treatment. For quantification of neuronal viability, cocultures exposed to TSZ after 4.5 hr after were stained using a propidium iodide-based (PI) cell viability kit (Biotium) according to manufacturer protocols. Cells were imaged live in Tyrode's buffer supplemented with Trolox. A total of four confocal images were taken randomly from each quadrant of the coverslip (top, bottom, left, right) and a corresponding bright field image was acquired to select neurons for quantification. The % cell death was determined by plotting a PI intensity curve for mock-treated WT neurons (TSZ with 0 ng TNF-α). A positive PI signal more than 2 S.D. from the mean was considered a positive signal for cell death. All Annexin-V-positive apoptotic neurons from the treatment were excluded from quantification (Annexin V positive neurons represented less than 3% of all cells, data not shown).

## FRAP

All cocultures in FRAP experiments were performed in Tyrode's buffer (140 mM NaCl; 10 mM HEPES pH 7.3; 5 mM KCl; 3 mM CaCl$_2$; 0.1 mM MgCl$_2$; 10 mM glucose; 10 nM Trolox pH 7.35) and supplemented with Trolox to reduce phototoxicity. Cells cultured in glass bottom Nunc chambers were placed on a heated stage in a humidified chamber setup and allowed to equilibrate prior to imaging.

The region of interest (ROI) for photobleaching were either identified based either on GFP expression or from Hoechst dye incubated at the start of experiments to identify the nucleus. For FRAP, cells were photobleached using a 405 nm point-scanning laser at 100% with a scan speed setting of 9. Photobleaching and imaging parameters were tested for individual GFP constructs to ensure optimal photobleaching with minimal cellular damage. All time-lapse images were acquired using a 40X1.3 NA oil objective outfitted with a heated collar at either 1 frame/s or 2 frames/min depending on experimental needs. Average baseline fluorescence values were obtained with a minimum of three consecutive images prior to the start of photobleaching. To prevent drift, the autofocus function was employed on GFP fluorescence (medium sensitivity). To ensure consistency in data collection, all imaging session were staggered with WT and *App* KI neurons to prevent any intersession imaging artefacts.

## Image acquisition

Unless otherwise stated, all confocal images, whether static or live cell imaging, were acquired using a Carl Zeiss LSM800 inverted scanning confocal microscope with either a 40X1.3 NA or 63 X/1.4NA Plan/Apochromat oil objectives. For individual replicates, imaging parameters were first optimized to capture the dynamic range of fluorescence emission across all samples which is then applied through the imaging session. For ICC, individual neurons were imaged as a z-stack with optical planes spaced at 0.45 μm apart using MAP2 and Hoechst as a guide for random selection of cells and optical planes for imaging. Neurons were equally sampled across four quadrants of the coverslip to ensure equal distribution of selected neurons. For IHC of brain sections, z-stack confocal micrographs were obtained from multiple sections from the CA1 pyramidal cell layer in the dorsal hippocampus. For high resolution imaging of individual NPC puncta, neurons were imaged with a with a 63 X, 1.4 NA oil DIC Plan Apochromatic objective on the Zeiss LSM800 AiryScan with a GrASP-PMT detector array. All images were processed post-acquisition using the AiryScan linear deconvolution algorithm. Whole forebrain sections were taken using a 20 X objective on the Zeiss AxioScanner.

## Image analysis and quantifications

All confocal image quantifications excluding time-lapse imaging experiments, were analysed using IMARIS versions 9.6.0–9.9.0. For surface reconstructions, 3D surfaces were rendered based on the fluorescence channel of interest with a 15% intensity threshold, surface smoothing, and 10 voxel exclusions. For quantification of signals in the cytoplasm, surfaces were created based on MAP2 staining excluding dendrites. For nuclear intensity, surfaces were created based on Hoechst stain. Surfaces were used to obtain mean intensity, volume (representative of coverage), and sphericity using the inbuild calculators. For fragmented surfaces, an average intensity across all surfaces was used. Percent change from WT was calculated and graphed to normalize for differences in imaging parameters between replicates. For quantification of Aβ, the average signal intensity in the soma is measured using 3D reconstruction of MAP2-positive neuronal soma. For quantification of brain sections, neurons were sampled from 3 to 4 micrographs taken per brain section. For each micrograph, five intact and non-overlapping neuronal nuclei were randomly sampled and quantified with IMARIS using a defined range of optical planes for reconstruction (0–10 μm z-plane for *App* KI and matched WT sections; 10–20 μm z-plane for APP/PS1 and matched WT sections) to limit variability between section. All imaging parameters, sample selection and quantification were performed consistently across animals in the experiment. For imaging of fluorescent dextrans in isolated nuclei, IMARIS surfaces were created based on Hoechst dye labeling after exclusion of out of focus nuclei. For AiryScan images, NPC puncta were defined with a Z-plane elongation correction (XY diameter 0.2 μm, Z diameter 0.4 μm) and a 25–30% quality threshold. IMARIS Spots was used to quantify the number of NPCs, and to perform nearest neighbor calculations. For time-lapse imaging, cells were analyzed using the Zeiss ZEN intensity measurements based on ROIs (region of interest). The nuclear ROI was determined based on a single snapshot of a Hoechst dye stain prior to time-lapse capture. All measurements were quantified as a % change from t=0 (T0; prebleached baseline). For all experiments, T0 is an average of 3 back-to-back frames of the baseline. For cytoplasmic GFP levels, a mosaic style ROI was constructed to cover the entire soma and averaged to generate a single track per cell.

## Statistical analysis

All data and graphs were plotted and analysed using GraphPad Prism 9.4.0. For experiments comparing WT and *App* KI, to ensure consistency between replicates, data obtained for each experiment were normalized against average WT or control values *per replicate* before statistical analysis were performed on the grouped data. This was done so that data for each replicate could be compared within the experiment, and to ensure that all imaging parameters that were optimized between replicates are normalized between sessions to prevent measurement artefacts. Where feasible, all individual replicates were tested and validated to demonstrate statistical significance. Unless otherwise stated, all bar graphs are reported as mean ± SEM. For all datasets, a normality test with Shapiro-Wilk was first performed with an alpha = 0.05 and a confidence interval of 95%. Depending on how the data is distributed, statistical significance for two-group comparisons were determined with Student's t-test or Mann-Whitney U test for parametric or non-parametric datasets. For more than two-group comparisons, a one-way ANOVA with Tukey's or Kruskal-Wallis post-hoc tests were performed for parametric and non-parametric datasets, respectively. For comparisons across two different variables a two-way ANOVA was used. A confidence interval of 95% was used and significance is represented with the p-value indications as follows: $p<0.05$(*), $p<0.01$ (**), $p<0.001$(***), $p<0.0001$ (****). In some cases, different notations are used besides the asterisks to denote specific comparisons which will be described in the figure legends. The complete statistical profile for each experiment can be found in *Supplementary file 1*.

## Acknowledgements

The authors acknowledge the contributions of Anna Barron, Christine Wong, Richard Reynolds, Suresh Jesuthasan, Albert Chen and Anusha Jayaraman for discussion and/or comments on the manuscript; Hiroki Sasaguri, Takaomi Saido and RIKEN BRC for usage of *App*<sup>NL-G-F/NL-G-F</sup> mice; Lye Yi Ming for providing technical support for the project, Anna Barron, Lauren Fairley, Albert Chen, and Gavin Dawe for reagents; Karen Chung for managing the mouse colony and George Augustine for spearheading the dementia research initiative. APP/PS1 brain sections were kind donations from Sheeja Navakhode and Sajikumar Sreedharan. This research is supported by Singapore Ministry of Education Academic Research Fund Tier 1 (MOE2018-T1-002-033), Academic Research Fund Tier 3 (MOE2017-T3-1-002) and Nanyang Technological University NAP Start-up funds.

## Additional information

### Funding

| Funder | Grant reference number | Author |
|---|---|---|
| Ministry of Education - Singapore | MOE2018-T1-002-033 | Jia Min Tan<br>Hui Rong Soon<br>Norliyana Zainolabidin<br>Toh Hean Ch'ng |
| Ministry of Education - Singapore | MOE2017- 21 T3-1-002 | Jia Min Tan<br>Hui Rong Soon<br>Norliyana Zainolabidin<br>Takaomi Saido<br>Toh Hean Ch'ng |

The funders had no role in study design, data collection and interpretation, or the decision to submit the work for publication.

### Author contributions

Vibhavari Aysha Bansal, Conceptualization, Data curation, Formal analysis, Supervision, Validation, Investigation, Visualization, Methodology, Writing – original draft, Project administration, Writing – review and editing; Jia Min Tan, Hui Rong Soon, Data curation, Formal analysis, Investigation, Methodology; Norliyana Zainolabidin, Formal analysis, Funding acquisition, Investigation, Methodology; Takaomi Saido, Resources; Toh Hean Ch'ng, Conceptualization, Resources, Data curation, Formal

analysis, Supervision, Funding acquisition, Validation, Investigation, Visualization, Methodology, Writing – original draft, Project administration, Writing – review and editing

**Author ORCIDs**
Vibhavari Aysha Bansal ⓘ https://orcid.org/0000-0002-9745-3633
Toh Hean Ch'ng ⓘ https://orcid.org/0000-0002-7170-1512

**Ethics**
All animal work was conducted in the Animal Research Facility (ARF) at Lee Kong Chian School of Medicine, Nanyang Technological University, Singapore, following protocols approved by Nanyang Technological University Institutional Animal Care and Use Committee (IACUC protocol A18091 and A18095).

Reviewer #2 (Public review): https://doi.org/10.7554/eLife.92069.3.sa1
Reviewer #3 (Public review): https://doi.org/10.7554/eLife.92069.3.sa2
Author response https://doi.org/10.7554/eLife.92069.3.sa3

# Additional files

**Supplementary files**
Supplementary file 1. Complete statistical output for experiments performed in the manuscript.
MDAR checklist

**Data availability**
All data generated and analyzed during this study are included in either the manuscript or in supporting supplementary files.

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
