## [Editor Report · eLife assessment]

This study focuses on nuclear pore complex dysfunction in a mouse model of Alzheimer's disease related Aβ pathology. If future revisions can adequately respond to the reviewer comments, the findings may eventually be **useful** in supporting the idea that nuclear cytoplasmic transport defects occur prior to plaque deposition in this disease model and may be caused by Alzheimer's disease pathology. However, even after revision, the work suffers from overinterpretation of some of the data and remains **incomplete** in several respects.

---

## [Referee Report · Reviewer #2 (Public review)]

Summary:

The authors try to establish that there is an Abeta-dependent loss of nuclear pores early in Alzheimer's disease. To do so the authors compared different NUP proteins and assessed their function by analyzing nuclear leakage and resistance to induction of nuclear damage and the associated necroptosis. The authors use a mouse knockin for hAPP with familial Alzheimer's mutations to model amyloidosis related to Alzheimer's disease. Treatment with an inhibitor of beta-amyloid production partially rescued the loss of nuclear pore proteins in young KI neurons, implicating beta-amyloid in Nuclear Pore dysfunction, a mechanism already described in other neurodegenerative diseases but not in Alzheimer's disease.

Comments on revised version:

Upon careful review, some of the critical concerns raised have yet to be fully addressed (the authors did not adequately address the two points of my public review or 5 of my 7 recommendation points), particularly regarding the effects of maturation stage or age. This has negatively impacted my initial enthusiasm for the paper, as the current approach does not fully capture the role of nuclear pore dysfunction in Alzheimer's disease, which is intimately dependent on aging. Here are specific recommendations for further revision:

(1) The manuscript would benefit from a clearer acknowledgement of the limitations concerning the effects of maturation or age. I recommend removing mentions of the effect of time, for example:

(i) Line 1 "4: "By using brain tissues and primary neurons cultured from App KI and wildtype (WT) mice, we observed a loss of NPCs in neuronal nuclei over time. "

(ii) Line 20 "13: "Similarly, in neuron cocultures, there was an 20 increase in intracellular Aβ levels over WT neurons that parallels the reduction of NUPs as neurons 21 mature from DIV "-28. "

(2) The subheading in the Discussion section, "Age-dependent decline in nuclear function during normal aging and in AD," could be more accurately retitled "Nuclear function decline" in AD" to avoid suggesting age dependence without the requisite data.

(3) Because primary neurons differentiate, mature, and age with time in culture, they are required to control for the developmental stage of your cultures. Please include the control data that would support cultures maturation stage, such as staining for axodendritic markers (e.g., MAP2), glial cell distribution (e.g., GFAP), and the balance of excitatory vs. inhibitory neuronal subpopulations (e.g., Gad65). This data is crucial for substantiating the culture conditions and the resulting interpretations.

---

## [Referee Report · Reviewer #3 (Public review)]

Summary:

This manuscript reports the novel observation of alterations in the nuclear pore (NUP) components and the function of the nuclear envelope in knock-in models of APP and presenilin mutations. The data show that loss of NUP immunoreactivity (IR) and pore density are observed at times prior to plaque deposition in this model. The loss of NUP IR is correlated with an increase in intraneuronal Abeta IR with two monoclonal antibodies that react with the N-terminus of Abeta. Similar results are observed in cultured neurons from APP-KI and Wt mice where further results with cultured neurons indicate that Abeta "drives" this process: incubation of neurons with oligomeric, but not monomeric or fibrillar Abeta causes loss of NUP IR, incubation with conditioned media from KI cells but not wt cells also causes loss of NUP IR and treatment with the gamma secretase inhibitor, NAPT partially blocks the loss of NUP IR. Further data show that nuclear envelope function is altered in KI cells and KI cells are more sensitive to TNFalpha-induced necroptosis. This is potentially an important and significant report, but how this fits within the larger picture of what is known about amyloid aggregation and accumulation and pathogenesis in neurons needs to be clarified. The results from mouse brains are strong, while the results from cultured cells are in some instances are of a lower magnitude, less convincing, ambiguous, and sometimes over-interpreted.

Comments on revised version:

I am disappointed in the responses submitted in the revised manuscript. Although there are two new supplemental figures shown, there is no new data that would be needed to address the points raised by myself and the other reviewers. For example, I asked the authors to provide data to place their observations on lower levels of NUPs and mislocalization of nuclear proteins in the context of previously published reports of nuclear amyloid pathology in APP mouse models reported by Pensalfini et al 2014 and Lee et al, 2022 who report amyloid fibrils in some neuronal nuclei along with rosettes of perinuclear autophagic vacuoles containing Abeta immunoreactive material that also stains with amyloid fibril-specific antibodies. In response the authors state: "We have devoted a section of the discussion to highlight some of these findings in the context of Pensalfini et al. 2014 and Lee et al. 2022. Lee et al. tested multiple animal strains to observe the Panthos structures but did not use the App KI mouse model. Since none of our experiments directly tested their observations (e.g. perinuclear fibrils or acidity of autophagic vesicles) in App KI, we decided to take a more conservative approach in our interpretations by framing the NPC deficits without specifying the nature of the intracellular Aβ. We note in discussion that it is entirely possible that App KI animals also show the same Panthos phenotypes and the perinuclear accumulation of Aβ which results in damaged NUPs. To do that, the Panthos phenotype must first be established in App KI mice. "

But the "discussion" is just a couple of sentences that misrepresents the findings of the previous publications and excuses for not doing experiments that the authors should do, like examining whether neurons with intranuclear amyloid and perinuclear autophagic vacuoles occur in the mouse model they use. They are experiments that they should do, and it would be easy to do. Is not an imposition to ask for this data because they presumably have the mouse brain tissue, so they could cut more brain sections and co-stain them with NUP antibodies and the antibodies against fibrillar Abeta and autophagic vesicle markers.

This is just one of many comments where new data is needed but not provided. Disappointing that the revised manuscript is not significantly improved.

---

## [Author Response]

The following is the authors’ response to the original reviews.

**Public Reviews:**
**Reviewer #1 (Public Review):**Weaknesses:However, the molecular mechanisms leading to NPC dysfunction and the cellular consequences of resulting compartmentalization defects are not as thoroughly explored. Results from complementary key experiments using western blot analysis are less impressive than microscopy data and do not show the same level of reduction. The antibodies recognizing multiple nucleoporins (RL1 and Mab414) could have been used to identify specific nucleoporins that are most affected, while the selection of Nup98 and Nup107 is not well explained.

The results for the Western blots are less impressive than single nuclei imaging analysis because the protocol for isolating brain nuclei is heterogeneous and includes non-neuronal cells. For this reason, we selected specific nucleoporins for Western blot studies to complement the nonspecificity of pan-NPC antibodies for which the detection is based on the glycosylated moieties. We reasoned that a combination of pan-NPC and select NUPs will give the strongest complementary validation for the mutant phenotype. We have discussed the rationale of NUP selection in discussion. In brief, we selected NUP107 as it is a major component of the Yscaffold complex and is a long-lived subunit of the NPCs (Boehmer et al., 2003; D'Angelo et al., 2009). NUP98 is a mobile nucleoporin and is associated with the central pore, nuclear basket and cytoplasmic filaments. Both NUPs have been implicated in degenerative disorders. (Eftekharzadeh et al., 2018; Wu et al., 2001).

There is also no clear hypothesis on how Aβ pathology may affect nucleoporin levels and NPC function. All functional NCT experiments are based on reporters or dyes, although one would expect widespread mislocalization of endogenous proteins, likely affecting many cellular pathways.We agree that the interaction between Aβ pathology and the NPC remains a work in progress. We decided to rigorously characterize Aβ-mediated deficits in App KI neurons – using different approaches and in more than one animal model – before moving on to explore mechanisms in subsequent studies, which we think deserves more extensive experiments. We seek your understanding and have included in the discussion, possible mechanisms for direct and indirect Aβ-mediated disruption of NPCs. We have also included an additional study to show the disruption in the localization of an endogenous nucleocytoplasmic protein – CRTC1 (cAMP Regulated Transcriptional Coactivator), which is CREB coactivator responsive to neural activity. We observed under basal and also in tetrodotoxin-silenced conditions, there is much higher CRTC1 in the nucleus in App KI neurons relative to WT. This reflects the compromised permeability barrier that we observed via FRAP studies. (Supplementary Figure S15).The second part of this manuscript reports that in App KI neurons, disruption in the permeability barrier and nucleocytoplasmic transport may enhance activation of key components of the necrosome complex that include receptor-interacting kinase 3 (RIPK3) and mixed lineage kinase domain1 like (MLKL) protein, resulting in an increase in TNFα-induced necroptosis. While this is of potential interest, it is not well integrated in the study. This potential disease pathway is not shown in the very simple schematic (Fig. 8) and is barely mentioned in the Discussion section, although it would deserve a more thorough examination.

The study of necroptosis is meant to showcase a single cellular pathway that requires nucleocytoplasmic transport for activation that is compromised and is relevant for AD. We agree there is much more to explore in this pathway but feel is outside the scope of this study. We have included a new illustration that models how damage to NPCs and permeability barrier results in enhanced vulnerability of App KI neurons for necroptosis (Supplemental figure S12).

**Reviewer #2 (Public Review):**
(1) Adding statistics and comparisons between wild-type changes at different times/ages to determine if the nuclear pore changes with time in wild-type neurons. The images show differences in the Nuclear pore in neurons from the wild-type mice, with time in culture and age. However, a rigorous statistical analysis is lacking to address the impact of age/development on NUP function. Although the authors state that nuclear pore transport is reported to be altered in normal brain aging, the authors either did not design their experiments to account for the normal aging mechanisms or overlooked the analysis of their data in this light.

All our quantifications and statistical comparisons in neuron cocultures are time-matched between WT and App KI neurons, and thus independent of age and maturity of the neurons in culture. The accelerated loss of NUP expression is evident across all time groups. However, we cannot compare across age groups in cultured neurons as the time-matched WT and App KI samples for each time point were processed and imaged separately as neurons matured over time (Fig. 1B-C). An experiment must be done simultaneously across all age groups to compare agerelated effects for WT and App KI neurons in order to account for time-dependent changes. Given the unique challenges of studying “aging” in culture systems, we opted to be more conservative in our interpretation of the results and as such, we were careful to describe the accelerated nuclear pore deficits in App KI neurons relative to time-matched WT expression and speculate its relationship to normal brain aging only in the discussion section. We seek your understanding in this matter. That said, we are able to capture the decline of the NPC in histology of brain sections and observed a statistically significant drop in WT NUP levels in animal sections across age groups where we quantified and compared the raw nuclear intensities from brain sections that were processed and imaged simultaneously across independent experiments (Fig. 1D-E). We have included a statement in the results section to highlight that point.

(2) Add experiments to assess the contribution of wild-type beta-amyloid accumulation with aging. It was described in 2012 (Guix FX, Wahle T, Vennekens K, Snellinx A, Chávez-Gutiérrez L, Ill-Raga G, Ramos-Fernandez E, Guardia-Laguarta C, Lleó A, Arimon M, Berezovska O, Muñoz FJ, Dotti CG, De Strooper B. 2012. Modification of γ-secretase by nitrosative stress links neuronal ageing to sporadic Alzheimer's disease. EMBO Mol Med 4:660-673, doi:10.1002/emmm.201200243) and 2021 (Burrinha T, Martinsson I, Gomes R, Terrasso AP, Gouras GK, Almeida CG. 2021. Upregulation of APP endocytosis by neuronal aging drives amyloid-dependent synapse loss. J Cell Sci 134. doi:10.1242/jcs.255752), 28 DIV neurons are senescent and accumulate beta-amyloid42. In addition, beta-amyloid 42 accumulates normally in the human brain (Baker-Nigh A, Vahedi S, Davis EG, Weintraub S, Bigio EH, Klein WL, GeulaC. 2015. Neuronal amyloid-β accumulation within cholinergic basal forebrain in ageing and Alzheimer's disease. Brain 138:1722-1737. doi:10.1093/brain/awv024), thus, it would be important to determine if it contributes to NUP dysfunction. Unfortunately, the authors tested the Abeta contribution at div14 when wild-type Abeta accumulation was undetected. It would enrich the paper and allow the authors to conclude about normal aging if additional experiments were performed, namely, treating 28Div neurons with DAPT and assessing if NUP is restored.

Your point is well-noted. We are intrigued at the potential contribution of WT Aβ to the decline in NUPs and NPC but decided to focus on mutant Aβ for this manuscript. We have observed negligible MOAB2-positive Aβ signals in WT neurons across all age groups (data not shown) but acknowledge the potential contributions of aging toward a reduction in NPC function. Instead, we have included a section in the discussion to highlight the aging-related expression of Aβ in WT neurons and a subset of the citations above to indicate a possible link with normal decay of NPCs.

**Reviewer #3 (Public Review):**
Weaknesses:(1) It does not consider the relationship of the findings here to other published work on the intraneuronal perinuclear and nuclear accumulation of amyloid in other transgenic mouse models and in humans.

We have updated the discussion to further elaborate on intraneuronal and perinuclear accumulation of amyloid and how that relates to our NPC phenotype.

(2) It appears to presume that soluble, secreted Abeta is responsible for the effect rather than the insoluble amyloid fibrils.

At present, our data cannot fully discount the role of fibrils or other forms of Aβ causing the NPC deficits, but our studies do show that external presence of Aβ (e.g. addition of synthetic oligomeric Aβ or App KI conditioned media) leads to intracellular accumulation and NPC dysfunction. We are aware that endogenous formation of fibrils could also contribute to the NPC dysfunction but refrained from drawing any conclusions without further studies. We have stated this in the discussion.

(5) It is not clear when the alteration in NUP expression begins in the KI mice as there is no time at which there is no difference between NUP expression in KI and Wt and the earliest time shown is 2 months. If NUP expression is decreased from the earliest times at birth, then this makes the significance of the observation of the association with amyloid pathology less clear.

The phenotype we observed early in neuronal cultures and in very young animals is subtle and in all our studies, the severity of the NUP phenotypes consistently correlates with elevated intracellular Aβ. We expect that by looking at earlier/younger neurons, the deficits will not be present. However, neurons before DIV7 are immature, and hence we chose not to include those in our observations. In animals, we observed Aβ expression in neuronal soma in young mice (2 mo.), but it is not clear when the deficits manifests and how early to look. While the NUP expression is reduced at an early stage, we speculate in discussion that cellular homeostatic mechanisms can compensate for any compromised nuclear functions and to maintain viability to the point where age-dependent degradation of cellular mechanisms will eventually lead to progression of AD.

**Reviewer #1 (Recommendations For The Authors):**
While the App KI model is suitable for modeling one key aspect of human AD, the use of the term "AD neurons" throughout the manuscript is misleading and should be avoided when describing experiments with "App KI neurons".

Noted and corrected.

The claim that Aβ pathology causes NPC dysfunction via reduced nucleoporin protein expression would be stronger if it was better supported by biochemical evidence based on western blots (WBs) to complement the strong microscopy data. The results shown in Figure 2H show a very weak effect compared to microscopy data that does not appear to match the quantification (e.g. Lamin-B1 staining appears reduced after 2 months in WB but not the graph). It is also not clear why nuclear fractionation is required. WB analyses with RL1 and MAB414 (that recognizes multiple FG-Nupsin ICCs and WBs) would help identify Nups that are most affected by Aβ pathology.

The weaker Western blot results is due to the heterogeneity of the nuclei we isolated from the whole brain which includes non-neuronal cells. We reasoned that isolating the nuclear fraction would give us a cleaner Western blot with fewer background bands as the input lysate is more specific. We also decided to use antibodies against specific NUPs as a way to complement the pan-NPC antibodies that detect glycosylation-enriched epitopes in the nucleus. We reasoned that Western blot identification of individual subunits should provide complementary and stronger evidence for the reduction of NUPs at the peptide level. Overall, we used four different nuclear pore antibodies (RL1, Mab414, NUP98, NUP107) to demonstrate the same mutant phenotype in App KI neurons.

While the observed NCT defects are discussed in detail, the authors do not present any potential mechanisms to be tested, how intracellular Aβ may impact NPCs. Does Aβ pathology affect nucleoporin expression or stability?

We have observed the presence of Aβ adjacent to the nuclear membrane and also in the cytosol via high resolution confocal microscopy (Supplementary Figure S14). Our primary goal in this paper is to provide convincing evidence – using different assays and in more than one mouse model – for the reduction of NUPs and lower NPC counts. We feel mechanistic details of Aβdriven NPC disruption requires more extensive experimentation more suitable for subsequent publications.

The very simple schematic just represents the loss of compartmentalization, without illustrating more complex concepts. It would also be improved by representing the outer and inner nuclear membrane fusing around the NPCs with a much wider perinuclear space between the membranes. As shown now, the nuclear envelope almost looks like a single membrane, while >60kDa proteins are shown at a similar size as the 125MDa NPC.

We have updated the illustration along with a new schematic for necroptosis (Supplementary Figure S12). We have refrained from giving specific details of the damage to the nuclear pore complex because it is not yet clear the nature of these deficits.

Misspelling of "Hoechst" as "Hochest" in several figures (Fig. 1, 2, S5, S7).

Noted and corrected

**Reviewer #2 (Recommendations For The Authors):**
(1) Additional data analysis is required concerning the wild-type controls. The figures show clear differences in the wild-type neurons with time in culture (referring to figures 1A, 1B, 1C; 2A, 2B, 2C, 2D,6E, 6F, 6G, s4) and in different ages (2E, 2F, 2G, 5B, 5C, 5D). The data analysis is shown for knockin vs the time-matched wild-type condition. The effect of time in wild-type neurons/mice should also be analyzed. All the data is suggested to be normalized to 7 DIV/2month wild-type neurons/mice. Were these experiments done with different time points of the same culture? This would be the best to conclude on the effect of time.

We have noted a decline of NUPs in WT neurons over time in primary cultures and in animal sections. This is not surprising since the NPC and nuclear signaling pathways deteriorate with age (Liu and Hetzer, 2022; Mertens et al., 2015). However, we are unable to do a direct comparison across age groups in cultured neurons as the time-matched WT and App KI neuronal samples for each time point were processed and imaged separately as neurons matured over time (Fig. 1B-C). Hence, we perform statistical analysis for each time-matched WT and App KI neurons. To be clear, multiple independent experiments across different cultures were performed at each time point. Given the inherent challenges of studying aging in culture systems, we opted to be more conservative in our interpretation of the results and as such, we were careful to describe the accelerated nuclear pore deficits in App KI neurons relative to WT levels without inferring the effect of time and speculate its relationship to normal brain aging only in the discussion section. That said, we are able to capture the decline of the nuclear pore complex across different age groups in histology of brain sections where we observed a drop in WT NUP levels in animal sections when we quantified and compared the raw nuclear intensities from brain sections that were processed and imaged simultaneously across independent experiments (Fig. 1D-E).

Similarly, in Figure 2H, why aren't 2 months compared with 14 months? Why were these ages chosen? 2 months is a young adult, and 14 months is a middle-aged adult. To conclude, aging should have included an age between 18 and 24 months old.

As with cultures, we isolated age-matched WT and App KI animals separately. We chose 2 to 14 months as they represent young and middle-aged adults as we wanted to showcase the nuclear pore deficits induced by the presence of Aβ without drawing a conclusion on the effects of age or time. That said, we do show histology of brain sections at 18 months of age with individual NUPs. We agree that the temporal aspects of NPC loss in WT neurons is interesting, however, given our experimental parameters, we cannot draw conclusions across different age groups at the moment.

In Figure 3, statistics between wild type should have been included.

Similar to the above comment, samples were processed and imaged independently across different groups, hence we cannot compare the datapoints across time.

(4) Additional quantification: The intensity of MOAB2 at 2 and 13 months should be measured as in Figure 3C.

Intracellular Aβ signal in 2-mo. old App KI mice is diffuse throughout the soma but in older animals, they are punctate. This observation was similarly described by Lord et al. for tgAPPArcSwe mice (Lord et al., 2006). We have included a confocal micrograph of MOAB-2 immunocytochemistry of a 13-mo. App KI brain section in supplemental figures (Supplementary Figure S13). We found it challenging to differentiate whether the signal is localized intracellularly or as an extracellular aggregate. Regardless, the differences in the quality and uneven distribution of Aβ signal makes any direct comparison of soma intensity across the different age groups harder to interpret in the context of the mutant phenotype.

(5) Additional experiments: Because primary neurons differentiate, mature, and age with time in culture, they are required to control for the developmental stage of your cultures. Analyzing neuronal markers such as doublecortin for neuronal precursors, MAP2 (or Tau) for dendritic/axonal maturation, synapsin for synaptic maturation, and accumulation of senescenceassociated beta-galactosidase (SA-Beta-Gal) as an aging marker.

As part of the maintenance of cultures, we stain cultures for axodendritic markers (e.g. MAP2), glial cell distribution (e.g GFAP) and excitatory vs. inhibitory neuronal subpopulations (e.g. Gad65) and synaptic markers (e.g. PSD95) to ensure that growth, survival and viability of neurons are not compromised (data not shown). These markers for maturity are routinely tracked to ensure proper development. We also test the health of the cultures (e.g. apoptosis, necrosis) and to look for cytoskeletal disruption or fragmentation for neuronal processes.

(6) Additional methods: The quantification of Abeta intensity in Figure 3 is not clearly explained in the methods. Was the intensity measured per field, per cell body?

The quantifications for Aβ are done for each MAP2-positive cell body and have included that statement in the methods.

(7) Missing in discussion integration and references to these papers:a. Mertens J, Paquola ACM, Ku M, Hatch E, Böhnke L, Ladjevardi S, McGrath S, Campbell B, Lee H, Herdy JR, Gonçalves JT, Toda T, Kim Y, Winkler J, Yao J, Hetzer MW, Gage FH. 2015. Directly Reprogrammed Human Neurons Retain Aging-Associated Transcriptomic Signatures and Reveal Age-Related Nucleocytoplasmic Defects. Cell Stem Cell 17:705-718. doi:10.1016/j.stem.2015.09.001b. Guix FX, Wahle T, Vennekens K, Snellinx A, Chávez-Gutiérrez L, Ill-Raga G, Ramos-Fernandez E, Guardia-Laguarta C, Lleó A, Arimon M, Berezovska O, Muñoz FJ, Dotti CG, De Strooper B. 2012. Modification of γ-secretase by nitrosative stress links neuronal ageing to sporadic Alzheimer's disease. EMBO Mol Med 4:660-673. doi:10.1002/emmm.201200243c. Burrinha T, Martinsson I, Gomes R, Terrasso AP, Gouras GK, Almeida CG. 2021. Upregulation of APP endocytosis by neuronal aging drives amyloid-dependent synapse loss. J Cell Sci 134. doi:10.1242/jcs.255752,

Neuronal amyloid-β accumulation within cholinergic basal forebrain in ageing and Alzheimer's disease. Brain 138:1722-1737. doi:10.1093/brain/awv024.

We have cited a subset of the papers in the discussion section and also expanded the discussion to include the possibility of time-dependent changes for Aβ expression in WT neurons.

**Reviewer #3 (Recommendations For The Authors):**
Specific comments:(1) Fig. 1D,E. Fig. 2E, F. This shows the change in NUP IR with time for the APP-KI, but there is also a difference between Wt and KI from the earliest time shown. How early is this difference apparent? From birth? The study should go back to the earliest time possible as the timing of the staining for NUP is important to correlate this with other events of intraneuronal Abeta and amyloid IR. Is the difference between 4 and 7-month ko mice in Figures 2G and 2F statistically significant? If not, perhaps we need a larger N to determine the timing accurately.

The point is well taken. We have not examined the WT and App KI brains before 2-mo. of age. At this early time point, the extracellular amyloid deposits are very low but intracellular Aβ can be readily detected in neuronal soma. We expect that as the animal ages, the Aβ inside cells will directly impact the NPC mutant phenotype, but it is unclear how early this phenotype manifests in animals and when we should look. To be clear, in less mature neurons (DIV7), the phenotype is very subtle and can only be observed via high resolution microscopy. The differences between 4-7 mo. old animals (Fig. 2F and G) in terms of severity of the reduction cannot be assessed as the age-matched animals for each time point were processed separately, but at each time point, we observed a significant reduction of NPC relative to WT. Nevertheless, in Figure 1E, we performed immunohistochemistry experiments with pan-NPC antibodies and quantified raw intensities to show a difference between 4/7-mo. with 13-mo. old animals.

(2) Similarly, the increase in Abeta IR is only shown for cultured neurons and only a single time point of 2 months is shown for CA1 in KI brain. Since a major point is that the decrease in NUP IR is correlated with an increase in Abeta IR, a more convincing approach would be to stain for both simultaneously in KI brain, especially since Abeta IR is quite sensitive to conformational variation between APP, Abeta, and aggregated forms and whether they are treated with denaturants for "antigen retrieval". The entire brain hemisphere should be shown as the pathology is not limited to CA1. There are many different Abeta antibodies that are specific to the amyloid state so it should be possible to come up with a set of antibodies and conditions that work for both Abeta and NUP staining.

The intracellular Aβ signal in 2-mo. old App KI mice is diffuse throughout the soma but in older animals, they are punctate. We have included a confocal micrograph of MOAB-2 immunocytochemistry of a 13-mo. App KI brain section (Supplementary Figure S13). We did not quantify Aβ as it was challenging to differentiate if the signal is intracellular Aβ or amyloid β plaques. Regardless, the differences in the quality and uneven distribution of Aβ signal makes any direct comparison of soma intensity across the different age groups much harder to interpret.

(3) Figure 3A. The staining with MOAB 2 and 82E1 appears qualitatively different with 82E1 exhibiting larger perinuclear puncta. Both antibodies appear to stain puncta inside the nucleus consistent with previously published reports of intranuclear amyloid IR. If these are flattened images, then 3D Z stacks should be shown to clarify this. Figure 3H shows what appears to be Abeta immunofluorescence quantitation in DAPT-treated cells, but the actual images are apparently not shown. The details of this experiment aren't clear or what antibody is used, but this may not be Abeta as many APP fragments that are not Abeta also react with antibodies like MOAB2.

Since 82E1 detects a larger epitope (aa1-16 as compared to 1-4 in MOAB-2), it is possible some forms of Aβ are differentially detected inside the cell. MOAB-2 is shown to detect the different forms of Aβ40 and 42, with a stronger selectivity for the latter. However, it is not known to react with APP or APP/CTFs (Youmans et al., 2012). DAPT-treated cells were processed and imaged as with other experiments in figure 3 using MOAB-2 antibodies to detect Aβ. We have included that information in the figure legends.

The way we image the cell is to collect LSM800 confocal stacks and use IMARIS software to render the nucleus in a 3D object prior to quantifying the intensity or coverage. In this way, we are capturing and quantifying the entire volume of the nucleus and not just a single plane. The majority of signal for MOAB-2 positive Aβ are punctate signals in the cytosol with a subset adjacent to the nucleus (Supplementary Figure 14; Airyscan; single plane). We also detected MOAB-2 signals coming from within the nucleus. The nature of this interaction between Aβ and the nuclear membrane/perinuclear space/nucleoplasm remains unclear.

(4) P20 L12. "We demonstrate an Aβ-driven loss of NUP expression in hippocampal neurons both in primary cocultures and in AD mouse models" It isn't clear that exogenous or extracellular Abeta drives this in living animals. All the data that demonstrate this is derived from cell culture and things may be very different (eg. Soluble Abeta concentration) in vivo. It is OK to speculate that the same thing happens in vivo, but to say it has been demonstrated in vivo is not correct.

We have rewritten the opening statement in the paragraph to narrowly define our observations in the context of App KI. We understand the caveats of our studies in primary cultures, but we have done our due diligence to study the phenomenon in different assays, using at least four different nuclear pore antibodies, and in more than one mouse model to show the deficits. We mentioned Aβ-driven loss but did not conclude which Aβ peptide (e.g. 40 vs. 42) or form (e.g. fibrillar) that drives the deficits. However, we have shown some data that oligomers and not monomers as well as extracellular Aβ can accumulate in the soma and trigger NPC deficits. We also state in the discussion that other possible mechanisms of action, mainly via indirect interactions of Aβ with the cell, could result in the deficits.

(5) P21, L21 "Inhibition of γ-secretase activity prevented cleavage of mutant APP and generation of Aβ, which led to the partial restoration of NUP levels". What the data actually shows is that treatment of the cells with DAPT led to partial restoration of NUP levels. Other studies have shown that DAPT is a gamma secretase inhibitor, so it is reasonable to suspect that the effect to gamma secretase activity, but the substrates and products are assumed rather than measured, so a little caution is a good idea here. For example, CTF alpha is also a substrate, producing P3, which is not considered abeta. The products Abeta and P3 also typically are secreted, where they can be further degraded. Abeta and P3 can also aggregate into amyloid, so whether the effect is really due to Abeta per se as a monomer or Abeta-containing aggregates isn't clear.

The point is noted. DAPT inhibition of γ-secretase can impact more than one substate as the complex can cleave multiple substrates. However, we have measured Aβ intensity which increases with DAPT, and while a singular experiment is insufficient to show direct Aβ involvement, we have performed other experiments that show a correlation of Aβ levels inside the soma and the degree of NPC reduction. This includes the direct application of synthetic Aβ42 oligomers. We agree the data cannot fully exclude the involvement of other γ-secretase cleavage products, but we feel there is strong enough evidence that Aβ – in whatever form - is at least partially if not, the main driver that promote these deficits.

(6) Discussion. The authors point to "intracellular Abeta" as a potential causative agent for decreased NUP expression and function and cite a number of papers reporting intracellular Abeta. (D'Andrea et al., 2001; Iulita et al., 2014; Kimura et al., 2003; LaFerla et al., 1997; Oddo et al., 2003b; Takahashi et al., 2004; Wirths et al., 2001). Most of these papers report immunoreactivity with Abeta antibodies and argue about whether this is really Abeta40 or 42 and not APP or APP-CTF immunoreactivity. What is missing from these papers and the discussion in this manuscript is that this is not just soluble Abeta, but Abeta amyloid of the same type that ends up in plaques because it has the same immunoreactivity with Abeta amyloid fibril-specific antibodies and even the classical anti-Abeta antibodies 6E10 and 4G8 after antigen retrieval as shown in papers by Pensalfini, et al., 2014 and Lee, et al., 2022 (1,2) who describe the evolution of neuritic plaques and their amyloid core beginning inside neurons. The term "dystrophic neurite" is a misnomer because the structures that resemble "neurites" morphologically are actually autophagic vesicles packed with Abeta and APP immunoreactive material which has the detergent insolubility properties of amyloid plaques. See (1,2). The apparent intranuclear IR of MOAB2 and 82E1 mentioned in comment 3 is relevant here. In Lee et al., the 3D serial section EM reconstruction of one of these neurons with perinuclear and nuclear amyloid shows abundant amyloid fibrils in the remnant of the nucleus. The nuclear envelope appears to break down as evidenced by the redistribution of NeuN immunoreactivity (Pensalfini et al.,) and other nuclear markers and the EM evidence (Lee et al.,). These papers are also improperly cited as evidence for a hypothetical intracellular source for soluble Abeta.

We have devoted a section of the discussion to highlight some of these findings in the context of Pensalfini et al. 2014 and Lee et al. 2022. Lee et al. tested multiple animal strains to observe the Panthos structures but did not use the App KI mouse model. Since none of our experiments directly tested their observations (e.g. perinuclear fibrils or acidity of autophagic vesicles) in App KI, we decided to take a more conservative approach in our interpretations by framing the NPC deficits without specifying the nature of the intracellular Aβ. We note in discussion that it is entirely possible that App KI animals also show the same Panthos phenotypes and the perinuclear accumulation of Aβ which results in damaged NUPs. To do that, the Panthos phenotype must first be established in App KI mice.

(7) The authors also cite the work of Ditaranto et al., 2001 and Ji et al., 2002 for Aβ-induced lysosomal leakage from these vesicular structures but overlook the original publications on Abeta-induced lysosomal leakage by Yang et al., (3) who further show that this is correlated with aggregation of Abeta42 upon internalization which also leads to the co-aggregation of APP and APP-CTFs in a detergent-insoluble form (4) and pulse-chase studies demonstrate that metabolically-labeled APP ultimately ends up as insoluble Abeta that have "ragged" N-termini (5). This work seems relevant to the results reported here as the perinuclear amyloid that the authors report here is likely to be the same insoluble, aggregated APP and APP-CTF-containing amyloid as that reported in references 1 and 2.

We have included the literature references in the discussion, highlighting the possibility of lysosomal leakage contributing to the NPC damage.

Minor points.(1) P2, L28 "permeability barrier facilities passive" should be 'facilitates'.(2) P7, L24 "homogenate and grounded for 5 additional strokes" One of the peculiarities of English is that the past tense of grind is ground. Grounded means something else.(3) P8, L9 "For synthetic Aβ experiments," Abeta what? 42? 40? It makes a difference and if it is Abeta42, you should be specific in the rest of the text where it is used.(4) P11, L14. "To determine if Aβ can trigger changes in nuclear structure and function" It seems a little early to start by presupposing that it is Abeta that triggers changes in nuclear structure and function. It sounds like you are starting out with a bias.(5) P11, L16,17 "While Aβ pathology is robustly detected in App KIs" At some point in the manuscript, either here or in the introduction, it would be useful to include a couple of sentences about what the pathology is in these mice along with the timing of the development of the pathology to compare with the results presented here. There are several types of amyloid deposits, "neuritic" plaques, diffuse plaques, and cerebrovascular amyloid. This is important because the early "neuritic" plaques are intraneuronal at least early on before the neuron dies. See (1,2).(6) P19, L10. "LMB is an inhibitor or CRM-1 mediated" should be of

All minor points have been addressed in the manuscript and figures.

References

(1) Pensalfini, A., Albay, R., 3rd, Rasool, S., Wu, J. W., Hatami, A., Arai, H., Margol, L., Milton, S., Poon, W. W., Corrada, M. M., Kawas, C. H., and Glabe, C. G. (2014) Intracellular amyloid and the neuronal origin of Alzheimer neuritic plaques. Neurobiol Dis 71C, 53-61(2) Lee, J. H., Yang, D. S., Goulbourne, C. N., Im, E., Stavrides, P., Pensalfini, A., Chan, H., Bouchet-Marquis, C., Bleiwas, C., Berg, M. J., Huo, C., Peddy, J., Pawlik, M., Levy, E., Rao, M., Staufenbiel, M., and Nixon, R. A. (2022) Faulty autolysosome acidification in Alzheimer’s disease mouse models induces autophagic build-up of Abeta in neurons, yielding senile plaques. Nat Neurosci 25, 688-701(3) Yang, A. J., Chandswangbhuvana, D., Margol, L., and Glabe, C. G. (1998) Loss of endosomal/lysosmal membrane impermeability is an early event in amyloid Aß1-42 pathogenesis. J. Neurosci. Res. 52, 691-698(4) Yang, A. J., Knauer, M., Burdick, D. A., and Glabe, C. (1995) Intracellular A beta 1-42 aggregates stimulate the accumulation of stable, insoluble amyloidogenic fragments of the amyloid precursor protein in transfected cells. J Biol Chem 270, 14786-14792(5) Yang, A., Chandswangbhuvana, D., Shu, T., Henschen, A., and Glabe, C. G. (1999) Intracellular accumulation of insoluble, newly synthesized Aßn-42 in APP transfected cells that have been treated with Aß1-42. J. Biol. Chem. 274, 20650-20656

References

Boehmer, T., Enninga, J., Dales, S., Blobel, G., and Zhong, H. (2003). Depletion of a single nucleoporin, Nup107, prevents the assembly of a subset of nucleoporins into the nuclear pore complex. Proc Natl Acad Sci U S A 100, 981-985.

D'Angelo, M.A., Raices, M., Panowski, S.H., and Hetzer, M.W. (2009). Age-dependent deterioration of nuclear pore complexes causes a loss of nuclear integrity in postmitotic cells. Cell 136, 284-295.

Eftekharzadeh, B., Daigle, J.G., Kapinos, L.E., Coyne, A., Schiantarelli, J., Carlomagno, Y., Cook, C., Miller, S.J., Dujardin, S., Amaral, A.S., et al. (2018). Tau Protein Disrupts Nucleocytoplasmic Transport in Alzheimer's Disease. Neuron 99, 925-940 e927.

Liu, J., and Hetzer, M.W. (2022). Nuclear pore complex maintenance and implications for agerelated diseases. Trends Cell Biol 32, 216-227.

Lord, A., Kalimo, H., Eckman, C., Zhang, X.Q., Lannfelt, L., and Nilsson, L.N. (2006). The Arctic Alzheimer mutation facilitates early intraneuronal Abeta aggregation and senile plaque formation in transgenic mice. Neurobiol Aging 27, 67-77.

Mertens, J., Paquola, A.C., Ku, M., Hatch, E., Bohnke, L., Ladjevardi, S., McGrath, S., Campbell, B., Lee, H., Herdy, J.R., et al. (2015). Directly Reprogrammed Human Neurons Retain Aging-Associated Transcriptomic Signatures and Reveal Age-Related Nucleocytoplasmic Defects. Cell stem cell 17, 705-718.

Wu, X., Kasper, L.H., Mantcheva, R.T., Mantchev, G.T., Springett, M.J., and van Deursen, J.M. (2001). Disruption of the FG nucleoporin NUP98 causes selective changes in nuclear pore complex stoichiometry and function. Proc Natl Acad Sci U S A 98, 3191-3196.

Youmans, K.L., Tai, L.M., Kanekiyo, T., Stine, W.B., Jr., Michon, S.C., Nwabuisi-Heath, E., Manelli, A.M., Fu, Y., Riordan, S., Eimer, W.A., et al. (2012). Intraneuronal Abeta detection in 5xFAD mice by a new Abeta-specific antibody. Molecular neurodegeneration 7, 8.